# Integrating systemic and molecular levels to infer key drivers sustaining metabolic adaptations

Pedro de Atauri[1,2☯]*, Míriam Tarrado-Castellarnau[1,2☯], Josep Tarragó-Celada[1], Carles Foguet[1,2], Effrosyni Karakitsou[1], Josep Joan Centelles[1,2], Marta Cascante[1,2]*

**1** Department of Biochemistry and Molecular Biomedicine & Institute of Biomedicine of Universitat de Barcelona, Faculty of Biology, Universitat de Barcelona, Barcelona, Spain, **2** Centro de Investigación Biomédica en Red de Enfermedades Hepáticas y Digestivas (CIBEREHD) and Metabolomics node at Spanish National Bioinformatics Institute (INB-ISCIII-ES-ELIXIR), Instituto de Salud Carlos III (ISCIII), Madrid, Spain

☯ These authors contributed equally to this work.
* pde_atauri@ub.edu (PdA); martacascante@ub.edu (MC)

**Data Availability Statement:** All relevant data are within the manuscript and its Supporting Information files. Two Mathematica notebooks, two pdf with instructions for use and excel files for

## Abstract

Metabolic adaptations to complex perturbations, like the response to pharmacological treatments in multifactorial diseases such as cancer, can be described through measurements of part of the fluxes and concentrations at the systemic level and individual transporter and enzyme activities at the molecular level. In the framework of Metabolic Control Analysis (MCA), ensembles of linear constraints can be built integrating these measurements at both systemic and molecular levels, which are expressed as relative differences or changes produced in the metabolic adaptation. Here, combining MCA with Linear Programming, an efficient computational strategy is developed to infer additional non-measured changes at the molecular level that are required to satisfy these constraints. An application of this strategy is illustrated by using a set of fluxes, concentrations, and differentially expressed genes that characterize the response to cyclin-dependent kinases 4 and 6 inhibition in colon cancer cells. Decreases and increases in transporter and enzyme individual activities required to reprogram the measured changes in fluxes and concentrations are compared with down-regulated and up-regulated metabolic genes to unveil those that are key molecular drivers of the metabolic response.

## Author summary

Deciphering the essential events in the reprogramming of metabolic networks subjected to complex perturbations, including the response to pharmacological treatments in multifactorial diseases like cancer, is crucial for the design of efficient therapies. Yet, tools to infer the molecular drivers sustaining such metabolic responses remain elusive for large metabolic networks. Here we develop an efficient computational strategy that integrates measured changes at systemic and molecular levels and combines metabolic control

the generation of results in Figs 2, 4 and 5 are freely available on Zenodo at link: http://dx.doi.org/10.5281/zenodo.5081161.

**Funding:** MC was supported by Agència de Gestió d'Ajuts Universitaris i de Recerca (AGAUR; Generalitat de Catalunya) (2017SGR1033), Instituto de Salud Carlos III (Centro de Investigación Biomédica en Red de Enfermedades Hepáticas y Digestivas - CIBEREHD-CB17/04/00023, Instituto Nacional de Bioinformática - PT17/0009/0018), Ministerio de Economía y Competitividad (SAF2017-89673-R) and Ministerio de Ciencia e Innovación (PID2020-115051RB-I00) (Co-funded by the European Regional Development Fund - "Una manera de hacer Europa"). JTC acknowledges the support from the Ministerio de Educación y Formación Profesional (FPU14-05992). MC acknowledges the support received through the prize "ICREA Academia" for excellence in research, funded by ICREA foundation – Generalitat de Catalunya. The funders had no role in study design, data collection, and analysis, decision to publish, or preparation of the manuscript.

**Competing interests:** The authors have declared that no competing interests exist.

analysis with linear programming tools to infer key molecular drivers sustaining the metabolic adaptations to complex perturbations, such as an antitumoral drug therapy. The collective behavior is approximated using linear expressions where the adaptation of systemic concentrations and fluxes to a perturbation is described as a function of the molecular reprogramming of transport and enzyme activities. Starting from measured changes in fluxes and concentrations, we identify changes in the reprogramming of transporter and enzyme activities that are required to orchestrate the metabolic adaptation of colon cancer cells to a cell cycle inhibitor.

## Introduction

Metabolism is a structured network of metabolites connected by transporters and enzyme-catalyzed reactions. The onset of multifactorial diseases like cancer and their response to pharmacological treatments are associated with complex metabolic adaptations [1,2]. Such metabolic adaptations are responses to large perturbations and often involve metabolic reprogramming driven by multiple changes in transporter and enzyme activities. At the systemic level, variables such as metabolite concentrations or reaction fluxes (i.e., transport and reaction rates) depend on the system's collective behavior and are measurable using various experimental methods [3]. In particular, complete estimations of the distribution of reaction fluxes throughout a metabolic network can be achieved with metabolic flux analysis supported by stable isotope-resolved metabolomics (SIRM) techniques [4,5].

These systemic variables depend on variations at the molecular level, such as individual transporter and enzyme activities. Given a perturbation, mathematical models are used to describe the adaptation of systemic concentrations and fluxes as a function of reprogramming at the molecular level. There are multiple modeling approaches for integrating information at the systemic and molecular levels. On the one hand, when there is a lack of detailed information at the molecular level, the dependencies between systemic reaction fluxes can be explored by stoichiometric models [6]. These models rely on reaction stoichiometry constraints to find viable steady-state intracellular flux distributions. These are the constraints used in the integration of SIRM data to obtain quantitative estimations of flux distributions, although limited to small or medium-sized metabolic networks. Alternatively, stoichiometric models can be applied at the genome-scale coupled to various optimization methods and the integration of multiple layers omics data [7–9]. On the other hand, when there is enough information at the molecular level, the dependencies of concentrations, fluxes, and individual activities, among others, can be explored with kinetic models [9–11]. By integrating kinetic rate laws for reaction and transport processes in systems of time-dependent ordinary differential equations, kinetic models explicitly describe reaction fluxes as a function of metabolite concentrations and individual activities, enabling dynamic simulations of systemic concentrations and fluxes. There are different kinetic modeling frameworks, each providing advantages and limitations [12]. Unfortunately, they are often limited by the availability of kinetic data and by the fact that the cell environment is far from the ideal conditions that are assumed by most kinetic models [13,14]. Several frameworks have been developed in this context of uncertainty, including approximate rate laws, such as (log)-linear or power-law based on linear Taylor's approximation [11]. These strategies are valid in the proximity of a reference steady-state and usually are associated with Metabolic Control Analysis (MCA) [15–19] or the closely related Biochemical Systems Theory (BST) [20,21]. They provide the advantage of simplified formulations and are frequently used in different computational methodologies based on optimization [22–25] and

**Table 1. Sensitivity coefficients.**

| | | A | B | dependencies |
|---|---|---|---|---|
| conc. control coefficients | $C_{v_k}^{x_i} = \frac{v_{ko}}{x_{io}}\frac{dx_i}{dv_k} = \frac{d\log x_i}{d\log v_k}$ | $x_i$ | $v_k$ | $\sum_{k=1}^{m} C_{v_k}^{x_i} = 0$ and $\sum_{k=1}^{m} C_{v_k}^{J_j} = 1$ |
| flux control coefficients | $C_{v_k}^{J_j} = \frac{v_{ko}}{J_{jo}}\frac{dJ_j}{dv_k} = \frac{d\log J_j}{d\log v_k}$ | $J_j$ | $v_k$ | (summation theorems) |
| "metabolite" elasticities | $\varepsilon_{x_i}^{v_k} = \frac{x_{io}}{v_{ko}}\frac{\partial v_k}{\partial x_i} = \frac{\partial \log v_k}{\partial \log x_i}$ | $v_k$ | $x_i$ | $\sum_{k=1}^{m} C_{v_k}^{x_a} \times \varepsilon_{x_b}^{v_k} = 0$, $\sum_{k=1}^{m} C_{v_k}^{x_a} \times \varepsilon_{x_a}^{v_k} = -1$, (a≠b) and $\sum_{k=1}^{m} C_{v_k}^{J_j} \times \varepsilon_{x_i}^{v_k} = 0$ (connectivity theorems) (see Fig 1 to other flux and concentration stoichiometric dependencies) |
| conc. response coefficients | $R_p^{x_i} = \frac{p}{x_{io}}\frac{dx_i}{dp} = \frac{d\log x_i}{d\log p}$ | $x_i$ | $p$ | $R_p^{x_i} = \sum_{k=1}^{m} C_{v_k}^{x_i} \times \varepsilon_p^{v_k}$ |
| flux response coefficients | $R_p^{J_j} = \frac{p}{J_{jo}}\frac{dJ_j}{dp} = \frac{d\log J_j}{d\log p}$ | $J_j$ | $p$ | $R_p^{J_j} = \sum_{k=1}^{m} C_{v_k}^{J_j} \times \varepsilon_p^{v_k}$ |
| "parameter" elasticities | $\varepsilon_p^{v_k} = \frac{p}{v_{ko}}\frac{\partial v_k}{\partial p} = \frac{\partial \log v_k}{\partial \log p}$ | $v_k$ | $p$ | |

sampling [17,26–28]. The ultimate objective of all these methodologies is the extraction (i.e., inference, prediction, identification) of new information from sets of observations and assumptions, which constitute groups of constraints that must be satisfied.

Both MCA and BST can be equivalently applied using sensitivity coefficients to quantify the variations at systemic and molecular levels in response to system perturbations. Sensitivity coefficients are implicitly associated with kinetic models and most frequently explicitly derived from them. MCA has been defined using formulations that are slightly different [15,16,18,19,28], although equivalent, to those used in BST [21,29], and where the dependencies of sensitivity coefficients at the systemic and molecular levels are explicitly described. At the systemic level, sensitivity coefficients can formally be divided into control and response coefficients. They describe variations in metabolite concentrations and reaction fluxes in response to perturbations in transporter and enzyme individual activities (control coefficients) or, in general, to any other parameter $p$ (response coefficients). Analogously, at the molecular level, elasticities are described as variations in transporter and enzyme activities in response to perturbations in metabolite concentrations (metabolite elasticities) or, in general, to any other perturbation (parameter elasticities). In Table 1, formal definitions of these sensitivity coefficients and the dependencies among them are provided for a metabolic network with $n$ internal metabolites ($i = 1,\ldots,n$), and $m$ transport and transformation processes ($j = k = 1,\ldots,m$), where each $x_i$ describes a metabolite concentration, each $J_i$ the systemic reaction flux (rate) through a particular process, and each $v_k$ the transport or enzyme activity of a particular process.

Each sensitivity is a dimensionless coefficient, which measures the fractional change in some variable $A$ per fractional change in some parameter $B$ around a steady-state ($x_{io}, J_{ko} = v_{ko}$). In control and response coefficients, the variations in systemic concentrations and fluxes are the result of the collective adaptation of the entire system after a transient period of adjustment, which is indicated using total derivatives [19]. In contrast, regarding elasticities, the changes in transport or enzyme activities happen as isolated individual processes. A positive sign indicates that variations in $A$ and $B$ magnitudes follow the same direction, both decreasing or both increasing. A negative sign indicates that variations in $A$ and $B$ magnitudes follow opposite directions, one increasing and the other decreasing.

Starting from the definition of concentration and flux response coefficient ($R_p^J$, $R_p^x$),

$$R_p^{x_i} = \frac{p}{x_{i0}}\frac{dx_i[v_1[p],\cdots,v_m[p]]}{dp} \tag{1}$$

$$R_p^{J_j} = \frac{p}{J_{j0}}\frac{dJ_j[v_1[p],\cdots,v_m[p]]}{dp} \tag{2}$$

with a direct application of the chain rule, Eqs (1) and (2) can be expanded to express response coefficients as a function of special elasticities and control coefficients [30–33]:

$$R_p^{x_i} = \sum_{k=1}^{m} C_{v_k}^{x_i} \times \varepsilon_p^{v_k} = C_{v_1}^{x_i} \times \varepsilon_p^{v_1} + \cdots + C_{v_m}^{x_i} \times \varepsilon_p^{v_m} \tag{3}$$

$$R_p^{J_j} = \sum_{k=1}^{m} C_{v_k}^{J_j} \times \varepsilon_p^{v_k} = C_{v_1}^{J_j} \times \varepsilon_p^{v_1} + \cdots + C_{v_m}^{J_j} \times \varepsilon_p^{v_m} \tag{4}$$

Each control coefficient $(C_v^x, C_v^J)$ describes the variations in a concentration or flux to the perturbation of one particular activity. Altogether, the complete set of control coefficients provides a full description of the potential behavior when a unique activity is perturbed. In response to a perturbation, each response coefficient in Eq (3) and (4) is a function of all control coefficients weighted by parameter elasticities ($\varepsilon_p^v$). Each response coefficient describes the overall variation in a concentration or reaction flux in response to a perturbation in some parameter $p$ that can affect one or multiple activities simultaneously, i.e., including any perturbation leading to a complex metabolic response.

A variety of optimization methods have exploited the dependencies among steady-state concentrations, fluxes, and system parameters such as enzyme levels or variations of them. They can take advantage of the particular formulation of the rate laws used in time-dependent differential equations, such as (log)-linear in MCA [34–36] and S-system and Generalized Mass Action (GMA) in BST [22–25,37]. At steady-state, mass balances provide sufficient constraints to account for all dependencies. The potential behavior is fixed through variables, such as metabolite elasticities for MCA, or their equivalent kinetic orders with BST. In the (log)-linear formulation [34,35,38], reformulated for mixed-integer linear programming (MILP), logarithmic deviations of the metabolite concentrations and enzyme levels take the role of decision variables, together with binary decision variables, in mass-balance derived linear constraints. The objective was to determine which enzymes should be present at different levels, the extent of such changes, and the accompanying modifications in the regulatory structure that optimize metabolic outputs, such as metabolite production [34,35]. Also, looking for steady-state optimizations in the BST framework, mass-balance derived constraints have been applied using the S-system and GMA power-law formulations [22–25]. In GMA, a power-law for each separate reaction is used to describe each metabolite's mass balance. In contrast, in S-system, the mass balance for each metabolite is represented by two competing power-law functions, one resulting from the aggregation of the separate power-laws for synthesis and the other from the aggregation of the separate power-laws for consumption [20]. Fixing rate constants and kinetic orders, the advantage of S-system representation is that in logarithmic coordinates the steady-state mass balances are linear equations, enabling the use of linear optimization techniques following a linear programming (LP) / MILP form [23,37,39–43]. GMA does not allow for a linear reformulation. However, efficient optimization tasks have also been performed using alternative optimization methods taking advantage of the structural regularity of the GMA representation, such as those relying on geometric programming techniques [23,24,44,45]. Another alternative optimization strategy that takes advantage of GMA is to apply outer approximation algorithms that decompose the target problem into master MILPs and slave nonlinear programming problems [22,46,47].

The formulation of optimization strategies using linear constraints enables to solve them using LP efficiently while guaranteeing convergence to an optimum solution point (minimum or maximum) [7]. In the methodology proposed in this paper, by combining MCA and LP, required decreases and increases previously unknown are extracted from linear constraints involving continuous domains in the form of bounded (closed) intervals measuring the

differences in reaction fluxes, metabolite concentrations, and individual activities comparing the initial and final states during the adaptation to a metabolic perturbation. Subject to a set of predetermined experimental values in the form of some initially restricted domains, the feasible range of values for all domains (previously restricted or not) is determined by successively minimizing and then maximizing the value for each reaction. Those domains reduced to only negative or positive values will identify required decreases or increases to be satisfied, respectively. Among them, changes required at the molecular level in individual activities will identify molecular drivers required for the metabolic adaptation.

As proof of concept, we use two case studies based on previously published experimental data. First, a glycolysis-case study covering the upper glycolysis and the oxidative branch of the pentose phosphate pathway (PPP) [48,49]. Second, a more complex cancer-case study expanded to all central carbon metabolism, associated with a set of experimental measurements obtained in cultured human colon cancer cells (HCT116) following the inhibition of cyclin-dependent kinases 4 and 6 (CDK4/6) [50]. A complete kinetic model reconstructed from experimental data supports the first case study. This model is built by assuming a full and ideal description of the system behavior, and it is used as a "toy" model to illustrate the proposed methodology. Under more realistic experimental conditions, the second study is built in a context of uncertainty, with partial knowledge and associated with a complex metabolic adaptation to a large perturbation. This study provides an example of the adaptation to simultaneous and coordinated changes in multiple activities, supported by a SIRM-based description of the altered flux distribution, together with the measurements of some metabolite concentrations and the analysis of the differential gene expression. Gene expression analyses are commonly applied to identify metabolic drivers, and therefore potential vulnerabilities to be exploited as targets in drug therapies. Although changes in gene expression should be related to changes in individual activities of their encoded products, alternative "moonlighting" roles cannot be discarded [51]. Also, although individual activities can be directly associated with enzyme concentrations measurable by specific activities, it is worth noting that they are also dependent on other measurable events, such as inactivation/activation by phosphorylation/dephosphorylation. Evidence supporting the role as a key molecular driver of an altered metabolic gene could be provided by the correspondence in the changes in transcript levels and the changes in the predicted activity of the encoded product, the latter being imposed by the changes observed at the systemic level. With the cancer-case study, we illustrate the application of the proposed methodology identifying such metabolic drivers by comparing changes in gene expression and changes required in the transporter and enzyme activities identified by the combination of MCA and LP.

## Results

### Response to a large perturbation

Response coefficients and special elasticities measure the response to infinitesimal (or small) perturbations around a particular steady-state. However, metabolic adaptations in response to the large perturbations will lead to complex changes in the qualitative steady-state leading to two separate states, before and after the perturbation. The control coefficients are redistributed during the adaptation from one state to the other. Taking a known flux control coefficient, assuming that variations in enzyme activity are small enough for this control coefficient to be constant, approximate predictions can be made about the change in the flux value by applying the following expression [16,21,52]:

$$\Delta \log J_j \approx C_{v_k}^{J_j} \times \Delta \log v_k \tag{5}$$

which corresponds to the integration of the definition of flux control coefficients in Table 1. Predictions for larger perturbations in enzyme activities can be made, although with an increasingly approximate value. Analogously, for perturbations involving several transporters and enzymes for fluxes and concentrations we expand the above expression to the following equations:

$$\Delta \log x_i = C_{v_1}^{x_i} \times \Delta \log v_1 + \cdots + C_{v_m}^{x_i} \times \Delta \log v_m = \sum_{k=1}^{m} C_{v_k}^{x_i} \times \Delta \log v_k \tag{6}$$

$$\Delta \log J_j = C_{v_1}^{J_j} \times \Delta \log v_1 + \cdots + C_{v_m}^{J_j} \times \Delta \log v_m = \sum_{k=1}^{m} C_{v_k}^{J_j} \times \Delta \log v_k \tag{7}$$

which are directly associated with Eqs (3) and (4), respectively, but constraining control coefficients with differences in reaction fluxes, metabolite concentrations, and individual activities. Thus, scaling by $\Delta \log p$, the two expressions are rewritten as:

$$\frac{\Delta \log x_i}{\Delta \log p} = C_{v_1}^{x_i} \times \frac{\Delta \log v_1}{\Delta \log p} + \cdots + C_{v_m}^{x_i} \times \frac{\Delta \log v_m}{\Delta \log p} = \sum_{k=1}^{m} C_{v_k}^{x_i} \times \frac{\Delta \log v_k}{\Delta \log p} \tag{8}$$

$$\frac{\Delta \log J_j}{\Delta \log p} = C_{v_1}^{J_j} \times \frac{\Delta \log v_1}{\Delta \log p} + \cdots + C_{v_m}^{J_j} \times \frac{\Delta \log v_m}{\Delta \log p} = \sum_{k=1}^{m} C_{v_k}^{J_j} \times \frac{\Delta \log v_k}{\Delta \log p} \tag{9}$$

where $\Delta \log x_i$, $\Delta \log J_j$, and $\Delta \log v_k$ are differences between the measurements before and after perturbation. The above expressions for a large perturbation are an approximation of the description of response coefficients as a function of control coefficients and special elasticities in Eqs (3) and (4), being equivalent for an infinitesimal perturbation. Accordingly, response coefficients and special elasticities could be approximated for large perturbations as:

$$R_p^{x_i} \approx \frac{\Delta \log x_i}{\Delta \log p}; R_p^{J_j} \approx \frac{\Delta \log J_j}{\Delta \log p}; \varepsilon_p^{v_k} \approx \frac{\Delta \log v_k}{\Delta \log p} \tag{10}$$

## Inference by bound contraction

Given a mathematical model that constrains the values of variables, such as concentrations, fluxes, and individual activities, this model must allow for the description of the different experimental situations that may potentially occur. Once the model is defined, it can be applied by adjusting the variables to reproduce particular conditions. In MCA, in the context of Eqs (3) and (4), or Eqs (6) and (7) for larger changes, the first level of "model description" can be provided by using control coefficients with fixed values. Once the model is established, a second level of "model application" can be done by adjusting the fluxes, concentrations, and individual activities. These variables can be taken as continuous domains in the form of bounded (closed) intervals measuring the differences in reaction fluxes, metabolite concentrations, and individual activities that are comparing the values before and after the adaptation to a metabolic perturbation. Because any combination of values for the variables must satisfy all model constraints, the restriction of the range of possible values for a part of the variables can, in turn, serve to infer new information in the form of additional restrictions on the values of any of the model variables. Starting with Eqs (6) and (7), an optimization-based procedure for bound contraction is reformulated following LP, where each variable is successively minimized and then maximized. In this reformulation, differences in fluxes, concentrations, and individual activities can be taken as decision variables in linear constraints, with control coefficients taken as the constant coefficients, fixed according to the available information. Therefore, given,

a. a complete description of the potential behavior, in the form of known control coefficients,

b. a metabolic response (adaptation), expressed using continuous domains for all differences in fluxes, concentrations, and individual activities; with generic lower and upper limits for all of them, and subject to additionally restricted lower and upper bounds for some of them based on experimental measurements,

and then, redistributing the terms in Eqs (6) and (7), we can define linear systems of equations with the following equalities,

$$
\begin{cases}
-\Delta \log x_i + \sum_{k=1}^{m} C_{v_k}^{x_i} \times \Delta \log v_k = 0 \\
-\Delta \log J_j + \sum_{k=1}^{m} C_{v_k}^{J_j} \times \Delta \log v_k = 0
\end{cases}
\tag{11}
$$

where, control coefficients are defined as constant parameters, and differences in fluxes, concentrations, and individual activities are defined as variables, for which initial lower (*lb*) and upper bounds (*ub*) of the initial domains are set in the form of inequalities ($lb_{x_i}^{in} \leq \Delta \log x_i \leq ub_{x_i}^{in}, lb_{J_j}^{in} \leq \Delta \log J_j \leq ub_{J_j}^{in}, lb_{v_k}^{in} \leq \Delta \log v_k \leq ub_{v_k}^{in}$). Taking all variables as decision variables, a series of LP problems can be solved. Each problem is solvable if all the model constraints and initial domains are satisfied, in other words, if all initial constraints are compatible. If the problem is solvable, each initial domain should contain at least one value satisfying all inequalities and constraints in Eq (11). Otherwise, the complete problem formulation must be discarded. The following maximization / minimization LP formulation is solved for each variable, one at a time:

maximize (and minimize) $\quad z = \Delta \log x_i, \Delta \log J_j, \Delta \log v_k \qquad \forall i \in N, \forall j, k \in M$

subject to

$$
\begin{aligned}
&-\Delta \log x_i + \sum_{k=1}^{m} C_{v_k}^{x_i} \times \Delta \log v_k = 0 \quad && \forall i \\
&-\Delta \log J_j + \sum_{k=1}^{m} C_{v_k}^{J_j} \times \Delta \log v_k = 0 \quad && \forall j \\
&lb_{x_i}^{in} \leq \Delta \log x_i \leq ub_{x_i}^{in} \quad && \forall i \\
&lb_{J_j}^{in} \leq \Delta \log J_j \leq ub_{J_j}^{in} \quad && \forall j \\
N = 1, \cdots, n \qquad &lb_{v_k}^{in} \leq \Delta \log v_k \leq ub_{v_k}^{in} \quad && \forall k \\
M = 1, \cdots, m \qquad &\Delta \log x_i, \Delta \log J_j, \Delta \log v_k \in \mathbb{R}
\end{aligned}
\tag{12}
$$

where any logarithm base would lead to the same results. The analysis is repeated for each decision variable ($\Delta \log J_j, \Delta \log x_i, \Delta \log v_k$); therefore, a total of $2n+4m$ LP analyses are performed to solve the complete problem. Each application of LP returns a combination of values for the complete set of decision variables that satisfies at the same time the set of constraints and inequalities in the initial domains, including the lower or upper bound of the minimized or maximized decision variable. Starting with an initial domain, taking the minimum and maximum solutions, this LP formulation provides a reduced final domain for each decision variable. Therefore, applying LP twice per each sensitivity coefficient will provide final lower and upper inequalities satisfying all conditions

($lb_{x_i}^{fi} \leq \Delta \log x_i \leq ub_{x_i}^{fi}, lb_{J_j}^{fi} \leq \Delta \log J_j \leq ub_{J_j}^{fi}, lb_{v_k}^{fi} \leq \Delta \log v_k \leq ub_{v_k}^{fi}$).

Previously, in the context of GMA-based applications of the outer approximation algorithm to analyze stress responses in yeast, a bound-contraction strategy was systematically applied [22,46,47]. The objective was to identify parameter regions in enzyme levels containing admissible solutions, and therefore changes, that were compatible with the considered physiological

constraints. Our use of LP is equivalent to that done in the framework of these stoichiometric models (Flux Variability Analysis) [53], where the mass balance around each metabolite is a system of linear constraints involving reaction fluxes. Subject to predetermined experimental values for a few fluxes, the feasible range of flux values is determined by minimizing and then maximizing the value for each reaction [53–58]. As the problem formulation is based on line-arization, our objective can be qualitative but not quantitative. Accordingly, the objective was to obtain collections of negative and positive signs, looking to capture the trends of the changes, decreases (negative) or increases (positive), and whether these trends are significant. Fixed-sign final domains that are required to explain all the initial constraints will be identified from domains that have only negative or only positive values. Although each final domain identifies a range of values that is required, it does not imply that, in turn, this domain is suffi-cient alone to constraint the initial domains. Therefore, in the context of the metabolic adapta-tions, signs will identify changes that are required, although not necessarily sufficient, to sustain all the initial constraints.

## Flow chart of the proposed analysis

Starting with the control coefficients calculated in Fig 1, Fig 2 illustrates the application of the formulation presented above using the model of the glycolysis-case study as a toy model. In this figure a flow chart is provided:

1. **Setting a model description**. Each problem formulation implies a model description in the form of a complete set of constant control coefficients. For the estimation of control coeffi-cients, different related matrix formulations have been developed in the context of MCA [31,32,59–62], which are closely related to those applied in the context of BST [20]. Accord-ingly, the complete set of constant control coefficients is usually presented as a matrix, where each element is a control coefficient. Setting the values of all metabolite elasticities and the steady-state ratios of dependent fluxes and concentrations, the system's potential behavior is fixed and described in the form of known control coefficients. These matrix methods are formulations that imply all summation and connectivity dependencies in Table 1, together with the stoichiometric dependencies of fluxes and concentrations of spe-cies involved in moiety conserved cycles. The dependencies among control coefficients in the panel A of Fig 2 were a consequence of these flux and concentration stoichiometric dependencies for the glycolysis-case in the panel C of Fig 1. A detailed description of this case-study and a detailed explanation of the calculation of control coefficients is provided in Material and Methods and Fig 1.

2. **Setting initial domains.** The variables are continuous domains measuring the differences in reaction fluxes, metabolite concentrations, and individual activities. **First**, common lower and upper bounds are set for the domains of all variables to be enclosed between a minimum lower bound and a maximum upper bound, assuming that differences outside this enclosure are not accepted. Although this enclosure is arbitrarily set, it can be adapted depending on the observed changes and contributes to reducing the space of solutions. In both case studies (initial domains in panel C in Fig 2 for glycolysis-case), common lower and upper bounds were set to be -3 and +3, which was consistent with the magnitude of expected changes. **Second**, the domains for differences in concentrations, fluxes, and indi-vidual activities measured experimentally are additionally restricted using the measured confidence intervals as bounds. These additional restrictions are a fundamental part of the problem formulation because the objective is to see the effect of these additional restrictions on all other variables, to see if new previously unknown fixed-sign domains finally appear.

**A**

Glc $R_6$ (NADPase)

(HK) $R_1$ $x_4$ (NADP) $x_5$ (NADPH)

(G6P) $x_1$ $R_5$

(G6PD)

(GPI) $R_2$

(F6P) $x_2$ $x_4 + x_5 = constant$

(PFK) $R_3$ **moiety conservation**

(FbP) $x_3$

(ALD) $R_4$

**B**

$$v_1 = V \frac{Glc}{K_m + Glc\left(1 + \frac{x_1}{K_i}\right)}$$

$Glc = 10 \, mM, V = 16.38 \, mM \, min^{-1},$
$K_m = 0.4019 \, mM, K_i = 0.111 mM$

$$v_2 = \frac{V_f \frac{x_1}{K_{m,1}} - Vr \frac{x_2}{K_{m,2}}}{1 + \frac{x_1}{K_{m,1}} + \frac{x_2}{K_{m,2}}}$$

$V_f = 2918.78 \, mM \, min^{-1}, K_{m,1} = 0.48 \, mM,$
$V_r = 4241.05 \, mM \, min^{-1}, K_{m,2} = 0.272 \, mM$

$$v_3 = V \frac{x_2^h}{K_{0.5}^h + x_2^h}$$

$V = 101.557 \, mM \, min^{-1}, K_{0.5} = 0.061 \, mM, h = 1.4744$

$$v_4 = V \frac{x_3}{K_m + x_3}$$

$V = 1560 \, mM \, min^{-1}, K_m = 0.1297 \, mM$

$$v_5 = V \frac{x_1 x_4}{\begin{pmatrix} K_{i,4}K_{m,1} + K_{m,1}x_4 + \\ K_{m,4}x_1 + x_4 x_1 + \\ (K_{i,4}K_{m,1}/K_{i,5})x_5 + \\ (K_{m,4}/K_{i,5})x_1 x_5 \end{pmatrix}}$$

$V = 20.7474 \, mM \, min^{-1}, K_{i,4} = 0.009 \, mM, K_{m,1} = 0.036 \, mM,$
$K_{m,4} = 0.0048 \, mM, K_{i,5} = 0.0011 \, mM, \; x_4 + x_5 = 0.0012 \, mM$

$$v_6 = k x_5$$

$k = 5906.15 \, min^{-1}$

**C**

$$\begin{cases} dx_1/dt = v_1 - v_2 - v_5 \\ dx_2/dt = v_2 - v_3 \\ dx_3/dt = v_3 - v_4 \\ dx_4/dt = v_6 - v_5 \\ dx_5/dt = v_5 - v_6 \end{cases}$$

$$\begin{cases} 0 = J_1 - J_2 - J_5 \\ 0 = J_2 - J_3 \\ 0 = J_3 - J_4 \\ 0 = J_6 - J_5 \\ 0 = J_5 - J_6 \end{cases}$$

**steady state**

$$\begin{cases} J_2 = J_1 - J_5 \\ J_3 = J_1 - J_5 \\ J_4 = J_1 - J_5 \\ J_6 = J_5 \\ x_5 = ct - x_4 \end{cases}$$

$J_1 = 11.81 \, mM \cdot min^{-1}$
$J_5 = 1.18 \, mM \cdot min^{-1}$
$x_4/x_5 = 5$

**D**

$$\begin{bmatrix} C_{v_2}^{J_1} & C_{v_3}^{J_1} & C_{v_4}^{J_1} & C_{v_6}^{J_1} & C_{v_1}^{J_1} & C_{v_5}^{J_1} \\ C_{v_2}^{J_5} & C_{v_3}^{J_5} & C_{v_4}^{J_5} & C_{v_6}^{J_5} & C_{v_1}^{J_5} & C_{v_5}^{J_5} \\ C_{v_2}^{x_4} & C_{v_3}^{x_4} & C_{v_4}^{x_4} & C_{v_6}^{x_4} & C_{v_1}^{x_4} & C_{v_5}^{x_4} \\ C_{v_2}^{x_1} & C_{v_3}^{x_1} & C_{v_4}^{x_1} & C_{v_6}^{x_1} & C_{v_1}^{x_1} & C_{v_5}^{x_1} \\ C_{v_2}^{x_2} & C_{v_3}^{x_2} & C_{v_4}^{x_2} & C_{v_6}^{x_2} & C_{v_1}^{x_2} & C_{v_5}^{x_2} \\ C_{v_2}^{x_3} & C_{v_3}^{x_3} & C_{v_4}^{x_3} & C_{v_6}^{x_3} & C_{v_1}^{x_3} & C_{v_5}^{x_3} \end{bmatrix} = \begin{bmatrix} \frac{J_1}{J_2} & -\frac{J_5}{J_2} & -\left(\varepsilon_{x_4}^{v_2} - \frac{x_4}{x_5}\varepsilon_{x_5}^{v_2}\right) & -\varepsilon_{x_1}^{v_2} & -\varepsilon_{x_2}^{v_2} & -\varepsilon_{x_3}^{v_2} \\ \frac{J_1}{J_3} & -\frac{J_5}{J_3} & -\left(\varepsilon_{x_4}^{v_3} - \frac{x_4}{x_5}\varepsilon_{x_5}^{v_3}\right) & -\varepsilon_{x_1}^{v_3} & -\varepsilon_{x_2}^{v_3} & -\varepsilon_{x_3}^{v_3} \\ \frac{J_1}{J_4} & -\frac{J_5}{J_4} & -\left(\varepsilon_{x_4}^{v_4} - \frac{x_4}{x_5}\varepsilon_{x_5}^{v_4}\right) & -\varepsilon_{x_1}^{v_4} & -\varepsilon_{x_2}^{v_4} & -\varepsilon_{x_3}^{v_4} \\ 0 & \frac{J_5}{J_6} & -\left(\varepsilon_{x_4}^{v_6} - \frac{x_4}{x_5}\varepsilon_{x_5}^{v_6}\right) & -\varepsilon_{x_1}^{v_6} & -\varepsilon_{x_2}^{v_6} & -\varepsilon_{x_3}^{v_6} \\ 1 & 0 & -\left(\varepsilon_{x_4}^{v_1} - \frac{x_4}{x_5}\varepsilon_{x_5}^{v_1}\right) & -\varepsilon_{x_1}^{v_1} & -\varepsilon_{x_2}^{v_1} & -\varepsilon_{x_3}^{v_1} \\ 0 & 1 & -\left(\varepsilon_{x_4}^{v_5} - \frac{x_4}{x_5}\varepsilon_{x_5}^{v_5}\right) & -\varepsilon_{x_1}^{v_5} & -\varepsilon_{x_2}^{v_5} & -\varepsilon_{x_3}^{v_5} \end{bmatrix}^{-1}$$

**Fig 1. Glycolysis-case study.** (A) Network scheme. (B) Rate laws and parameters. (C) System of ordinary differential equations (ODEs) and stoichiometric dependencies of fluxes and concentrations. (D) Calculation of control coefficients.

As an example, for the glycolysis-case the observed metabolic adaptation, expressed using additionally restricted domains (initial domains in red, panel C Fig 2), could be: "an increase in the flux through the first step ($J_1$) has been observed, although neither the concentrations of most of the intermediaries ($x_1$, $x_2$, $x_3$, $x_4$), nor the individual activity through one of the branches ($v_5$), have changed". As for the model description, each problem formulation implies a complete set of initial domains, with some of them additionally restricted.

3. **Solving final domains to identify negative and positive signs.** Once the problem has been formulated, the maximization / minimization LP formulation in Eq (12) is solved for each variable, one at a time. In the analyzed example (final domains in panel C in Fig 2), the problem was solvable; therefore, all initial constraints were compatible. The system was constrained enough to reduce the domain for most of the variables, even restricting some of them, fluxes and individual activities, to have fixed-sign domains with only positive values. As a measure of this bound contraction, a percentage gain metric has been added for each variable to quantify the percentage of reduction in the size of their final domains with respect to their initial domains. Looking at the molecular level, the required change in individual activities identifies four key molecular drivers ($v_1$, $v_2$, $v_3$, $v_4$) that are required, although not sufficient, to explain the whole set of constraints for the observed metabolic adaptation.

Starting with calculated control coefficients and initial values, a Mathematica notebook is provided to solve the final domains and identify negative and positive signs (see Calculations in Material and Methods).

Constraints propagate in all directions among systemic and molecular levels. However, as shown in previous works, the analyses of alternative topological designs under MCA [63] and BST [64] applying sampling techniques highlight the relevance of the structural constraints on the possible values of sensitivity coefficients. The control coefficients provide a complete description of the potential collective behavior, which implies not only summation and connectivity theorems but also all stoichiometric flux and concentration dependencies. As highlighted in panel C Fig 2 for the glycolysis-case (see labels a, b, and c), by setting the potential behavior with known control coefficients, all these dependencies for control coefficients in panel A are also implicit in the linear formulation for the differences in fluxes and concentrations.

The first case study was used to illustrate the application of the proposed analysis using a toy model. In contrast, the second case study is an example of a more complex metabolic response to a large perturbation. A detailed description of this second case-study is provided in the Material and Methods section and S1 and S2 Tables.

## Adding constraints among individual activities to the problem formulation

In addition to a larger number of metabolites and processes, the problem formulation can require additional constraints. In the formulation in Eq (12) alone, it is implicit that each individual activity corresponds to an independent variable. However, as happens in the cancer-case study, some activities can be interdependent, such as the two activities ($R_{13}$ and $R_{14}$) catalyzed by transketolase (TKT), or can be assumed to behave as a coordinated block. Thus, additional constraints were added in the problem formulation to include dependencies among

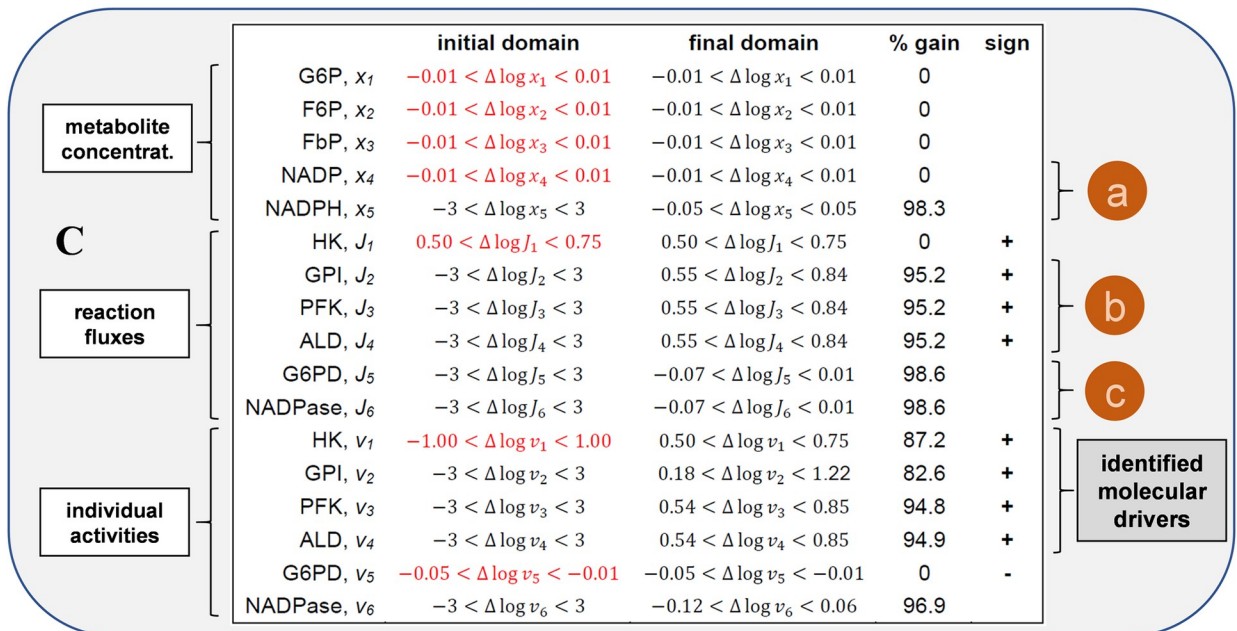

**A  Model description (constant control coefficients)**

$C_{v_2}^{J_1} = 0.01$   $C_{v_3}^{J_1} = 0.14$   $C_{v_4}^{J_1} = 0.00$   $C_{v_6}^{J_1} = 0.00$   $C_{v_1}^{J_1} = 0.83$   $C_{v_5}^{J_1} = 0.01$

$C_{v_3}^{J_5} = -0.02$   $C_{v_3}^{J_5} = -0.27$   $C_{v_4}^{J_5} = 0.00$   $C_{v_6}^{J_5} = 0.23$   $C_{v_1}^{J_5} = 0.32$   $C_{v_5}^{J_5} = 0.74$

$C_{v_2}^{x_4} = 0.00$   $C_{v_3}^{x_4} = 0.05$   $C_{v_4}^{x_4} = 0.00$   $C_{v_6}^{x_4} = 0.15$   $C_{v_1}^{x_4} = -0.06$   $C_{v_5}^{x_4} = -0.15$

$C_{v_2}^{x_1} = -0.04$   $C_{v_3}^{x_1} = -0.58$   $C_{v_4}^{x_1} = 0.00$   $C_{v_6}^{x_1} = -0.02$   $C_{v_1}^{x_1} = 0.69$   $C_{v_5}^{x_1} = -0.05$

$C_{v_2}^{x_2} = 0.01$   $C_{v_3}^{x_2} = -0.61$   $C_{v_4}^{x_2} = 0.00$   $C_{v_6}^{x_2} = -0.02$   $C_{v_1}^{x_2} = 0.67$   $C_{v_5}^{x_2} = -0.05$

$C_{v_2}^{x_3} = 0.01$   $C_{v_3}^{x_3} = 0.19$   $C_{v_4}^{x_3} = -1.01$   $C_{v_6}^{x_3} = -0.02$   $C_{v_1}^{x_3} = 0.89$   $C_{v_5}^{x_3} = -0.07$

**(a)** $\quad C_{v_i}^{x_5} = -\dfrac{x_4}{x_5} C_{v_i}^{x_4}$

**(b)** $\quad C_{v_i}^{J_q} = \dfrac{J_1}{J_q} C_{v_i}^{J_1} - \dfrac{J_5}{J_q} C_{v_i}^{J_5}$

$\quad\quad J_q = J_2 = J_3 = J_4 = J_1 - J_5$

**(c)** $\quad C_{v_i}^{J_6} = \dfrac{J_5}{J_6} C_{v_i}^{J_5}; J_5 = J_6$

**Stoichiometric flux and concentration dependencies**

**1**

**B**    **Model application (Bound contraction)**

maximize (and minimize)   $z = \Delta \log x_i, \Delta \log J_j, \Delta \log v_k \qquad \forall i \in N, \forall j, k \in M$

subject to

$$-\Delta \log x_i + \sum_{k=1}^{m} C_{v_k}^{x_i} \times \Delta \log v_k = 0 \quad \forall i$$

$$-\Delta \log J_j + \sum_{k=1}^{m} C_{v_k}^{J_j} \times \Delta \log v_k = 0 \quad \forall j$$

$$lb_{x_i}^{in} \leq \Delta \log x_i \leq ub_{x_i}^{in} \qquad \forall i$$

$$lb_{J_j}^{in} \leq \Delta \log J_j \leq ub_{J_j}^{in} \qquad \forall j$$

$N = 1, \cdots, n$   $\qquad lb_{v_k}^{in} \leq \Delta \log v_k \leq ub_{v_k}^{in} \qquad \forall k$

$M = 1, \cdots, m$   $\qquad \Delta \log x_i, \Delta \log J_j, \Delta \log v_k \in \mathbb{R}$

**2**    **3**

**C**

| | | initial domain | final domain | % gain | sign |
|---|---|---|---|---|---|
| **metabolite concentrat.** | G6P, $x_1$ | $-0.01 < \Delta \log x_1 < 0.01$ | $-0.01 < \Delta \log x_1 < 0.01$ | 0 | |
| | F6P, $x_2$ | $-0.01 < \Delta \log x_2 < 0.01$ | $-0.01 < \Delta \log x_2 < 0.01$ | 0 | |
| | FbP, $x_3$ | $-0.01 < \Delta \log x_3 < 0.01$ | $-0.01 < \Delta \log x_3 < 0.01$ | 0 | |
| | NADP, $x_4$ | $-0.01 < \Delta \log x_4 < 0.01$ | $-0.01 < \Delta \log x_4 < 0.01$ | 0 | |
| | NADPH, $x_5$ | $-3 < \Delta \log x_5 < 3$ | $-0.05 < \Delta \log x_5 < 0.05$ | 98.3 | |
| **reaction fluxes** | HK, $J_1$ | $0.50 < \Delta \log J_1 < 0.75$ | $0.50 < \Delta \log J_1 < 0.75$ | 0 | + |
| | GPI, $J_2$ | $-3 < \Delta \log J_2 < 3$ | $0.55 < \Delta \log J_2 < 0.84$ | 95.2 | + |
| | PFK, $J_3$ | $-3 < \Delta \log J_3 < 3$ | $0.55 < \Delta \log J_3 < 0.84$ | 95.2 | + |
| | ALD, $J_4$ | $-3 < \Delta \log J_4 < 3$ | $0.55 < \Delta \log J_4 < 0.84$ | 95.2 | + |
| | G6PD, $J_5$ | $-3 < \Delta \log J_5 < 3$ | $-0.07 < \Delta \log J_5 < 0.01$ | 98.6 | |
| | NADPase, $J_6$ | $-3 < \Delta \log J_6 < 3$ | $-0.07 < \Delta \log J_6 < 0.01$ | 98.6 | |
| **individual activities** | HK, $v_1$ | $-1.00 < \Delta \log v_1 < 1.00$ | $0.50 < \Delta \log v_1 < 0.75$ | 87.2 | + |
| | GPI, $v_2$ | $-3 < \Delta \log v_2 < 3$ | $0.18 < \Delta \log v_2 < 1.22$ | 82.6 | + |
| | PFK, $v_3$ | $-3 < \Delta \log v_3 < 3$ | $0.54 < \Delta \log v_3 < 0.85$ | 94.8 | + |
| | ALD, $v_4$ | $-3 < \Delta \log v_4 < 3$ | $0.54 < \Delta \log v_4 < 0.85$ | 94.9 | + |
| | G6PD, $v_5$ | $-0.05 < \Delta \log v_5 < -0.01$ | $-0.05 < \Delta \log v_5 < -0.01$ | 0 | - |
| | NADPase, $v_6$ | $-3 < \Delta \log v_6 < 3$ | $-0.12 < \Delta \log v_6 < 0.06$ | 96.9 | |

**a**   **b**   **c**   **identified molecular drivers**

**Fig 2. Flow chart of the proposed analysis.** (A) Model description in the form of fixed control coefficients. The values correspond to the glycolysis-case. Inside the brown square are the dependencies among control coefficients. (B) Maximization / minimization LP formulation for bound contraction in Eq (12). (C) Columns in Table: 1) *variable* (reaction flux, metabolite concentration or individual activity); 2) *initial domain*, described using inequality notation, with additionally (experimentally) restricted initial domains in red; 3) *final domain*, described using inequality notation; 4) *% gain*, comparing initial and final domains for each variable; and 5) *sign*, fixed positive signs (values can be only positive) or fixed negative signs (values can be only negative). (C) All logarithms are to base two. See Material and Methods for a supplementary description of the model and abbreviations.

transporter/enzyme activities. On the one hand, a constraint was added for the two activities for TKT:

$$\Delta \log v_{13} = \Delta \log v_{14} \qquad (13)$$

On the other hand, in the initial part of glycolysis, the consecutive activities for glucose (Glc) transport (Glc transp) ($R_{01}$) and the reactions catalyzed by hexokinase (HK) ($R_{02}$), phosphofructokinase (PFK) ($R_{04}$), and enolase (ENO) ($R_{08}$) were assumed to be coordinately regulated ($\Delta \log v_{01} = \Delta \log v_{02} = \Delta \log v_{04} = \Delta \log v_{08}$):

$$\begin{aligned} \Delta \log v_{01} &= \Delta \log v_{02} \\ \Delta \log v_{01} &= \Delta \log v_{04} \\ \Delta \log v_{01} &= \Delta \log v_{08} \end{aligned} \qquad (14)$$

## Setting a model description: coupling the problem formulation with uncertainty

In the glycolysis-case study, following a local perspective, a unique matrix of control coefficients was derived from a detailed kinetic model built around a well-defined steady-state. Therefore, by solving a unique set of linear constraints. In contrast, the cancer-case study was done under realistic experimental conditions, involving a more complex metabolic response to CDK4/6 inhibition, with two different steady-states, before and after CDK4/6 inhibition. On the one hand, a unique matrix of known and constant control coefficients cannot adequately be applied because the values change during the adaptation to the large perturbation. On the other hand, there is limited availability of data to estimate metabolite elasticities and the ratios among stoichiometrically dependent fluxes and concentrations. Moving from the local analysis, both limitations can be tackled simultaneously by applying a more global analysis, solving ensembles of the complete problem, each one associated with a matrix of control coefficients generated by random sampling methods, therefore covering a wider parameter space as described by Kent et al. [65]. In the context of the (MCA-based) (log)-linear formulation, control coefficients and response coefficients are derived under uncertainty by sampling techniques (ORACLE) [17,28,66,67], integrating data such as flux distributions and displacements of the reactions from equilibrium. When details about the enzyme's rate expressions are not available, elasticity values can be randomly generated. Also, stability analyses based on the analysis of Jacobian matrices can be derived from random sampling of elasticities (Structural kinetic modeling) [26,27,68]. We adapted these tools to our objective and data available in the cancer-case study. An ensemble of 100 problems was formulated, each associated with a complete set of control coefficients estimated by direct sampling of metabolite elasticities. Like the glycolysis-case study, metabolite elasticities are a function of the steady-state, transport or kinetic mechanisms, and regulatory states. Although we did not use a complete kinetic model accounting precisely for elasticities, together with measured fluxes, the sampling of elasticities was constrained by different assumptions, including enzyme saturation, displacement from

equilibrium, and literature-based data regarding moiety conservations, inhibitions, and activations.

See Material and Methods for a detailed explanation of the calculation of control coefficients coupled with the sampling of metabolite elasticities.

## Setting initial domains

The model used to characterize the metabolic response to CDK4/6 inhibition covered the central carbon metabolism, with a subset of the transport and reaction processes conforming a core network (Fig 3) that is wrapped in a simplified network of boundary processes. As discussed above, first, common lower and upper bounds were set to be -3 and +3. Second, the initial domains for differences in all the reaction fluxes ($lb^{in}_{J_j} \leq \Delta \log J_j \leq ub^{in}_{J_j}$) and some of the metabolite concentrations ($lb^{in}_{x_i} \leq \Delta \log x_i \leq ub^{in}_{x_i}$) were additionally restricted according to experimental observations. Also, part of the initial domains for differences in individual activities ($lb^{in}_{v_k} \leq \Delta \log v_k \leq ub^{in}_{v_k}$) were additionally restricted to force the adaptation of the lightly modeled boundaries to the CDK4/6 inhibition. Thus, the individual activities ($\Delta \log v_k$) for the simplified-boundary processes, that are a link of the core network with the whole cellular metabolic environment, were set to have the same tightly restricted initial domains as the corresponding reaction fluxes ($lb^{in}_{J_k} \leq \Delta \log v_k \leq ub^{in}_{J_k}$).

See S3 Table and Material and Methods for additional details regarding the initial domains.

## Solving final domains to identify negative and positive signs

Fig 4 illustrates the application of the developed strategy using the cancer-case study. Once initial values were set, the analysis was repeated over the ensemble of 100 formulated problems. Among these analyses, 16 were not compatible with all initial domains and constraints and were discarded. For each problem, we identified changes at the systemic and molecular levels with fixed-sign final domains and, therefore, associated with decreases (negative) or increases (positive) required to explain the observed differences. They included 14 individual activities, seven concentrations, and six reaction fluxes. The analysis of the unions and intersections in Fig 5 provides complementary information to that provided by the signs. Although the applied analysis has a qualitative value, such unions and intersections provide a numerical summary to assess the magnitude of these changes, which was significant in all the cases, and also (as the signs) tell us about the dependence on the sampling. A coincidence of the lower and upper bounds of the unions and the intersections will correspond to a no dependence on the sampling of metabolite elasticities.

The degree of identifiability of the signs depends on the degrees of freedom for each variable, and therefore, on the available information. A detailed local description of the variations in all fluxes and concentrations around a process will impose variations in the associated individual activity. As shown in Fig 4, signs identified for some of the variables were repeatedly negative or positive signs, thus independent of the sampled metabolite elasticities. The signs largely depended first on structural constraints [63,64], although a part of them was also dependent on metabolite elasticities. Among others, metabolite elasticities are a function of reversibility levels, which were fixed values in all the formulated problems (see Material and Methods for the cancer-case description). For example, the required increase in fumarate (Fum) concentration and glutaminase (GLS) activity disappear when the reactions $R_{25}$ (reversible hydration of Fum in malate (Mal)) and $R_{35}$ (transport of glutamine (Gln)), respectively, are switched to be far from equilibrium ($\rho = 0.9$ to $\rho = 0$). As another example of the dependence on metabolite elasticities, the constraints associated with the sampled elasticities can be

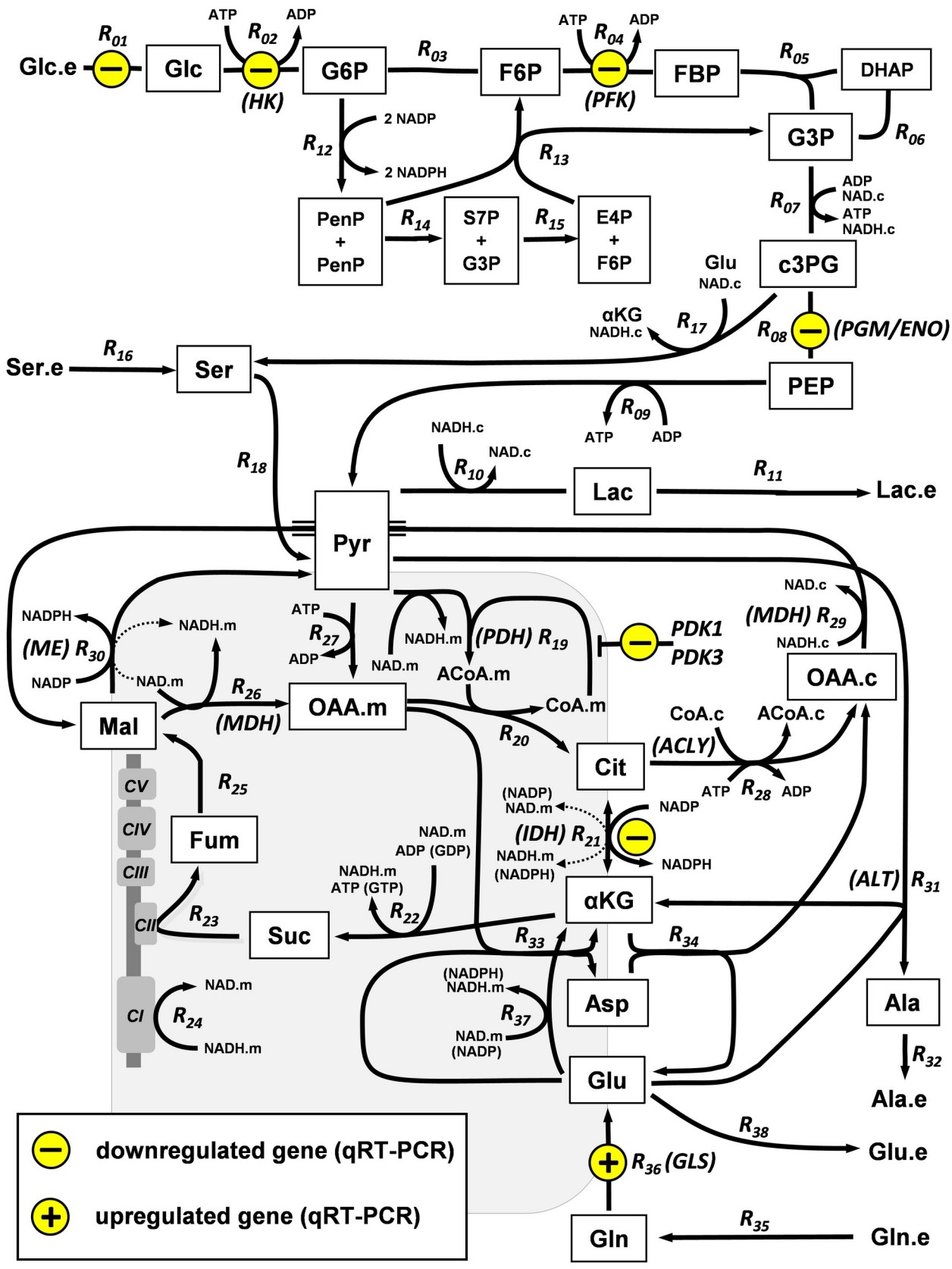

**Fig 3. Cancer-case study.** Scheme of the core network.

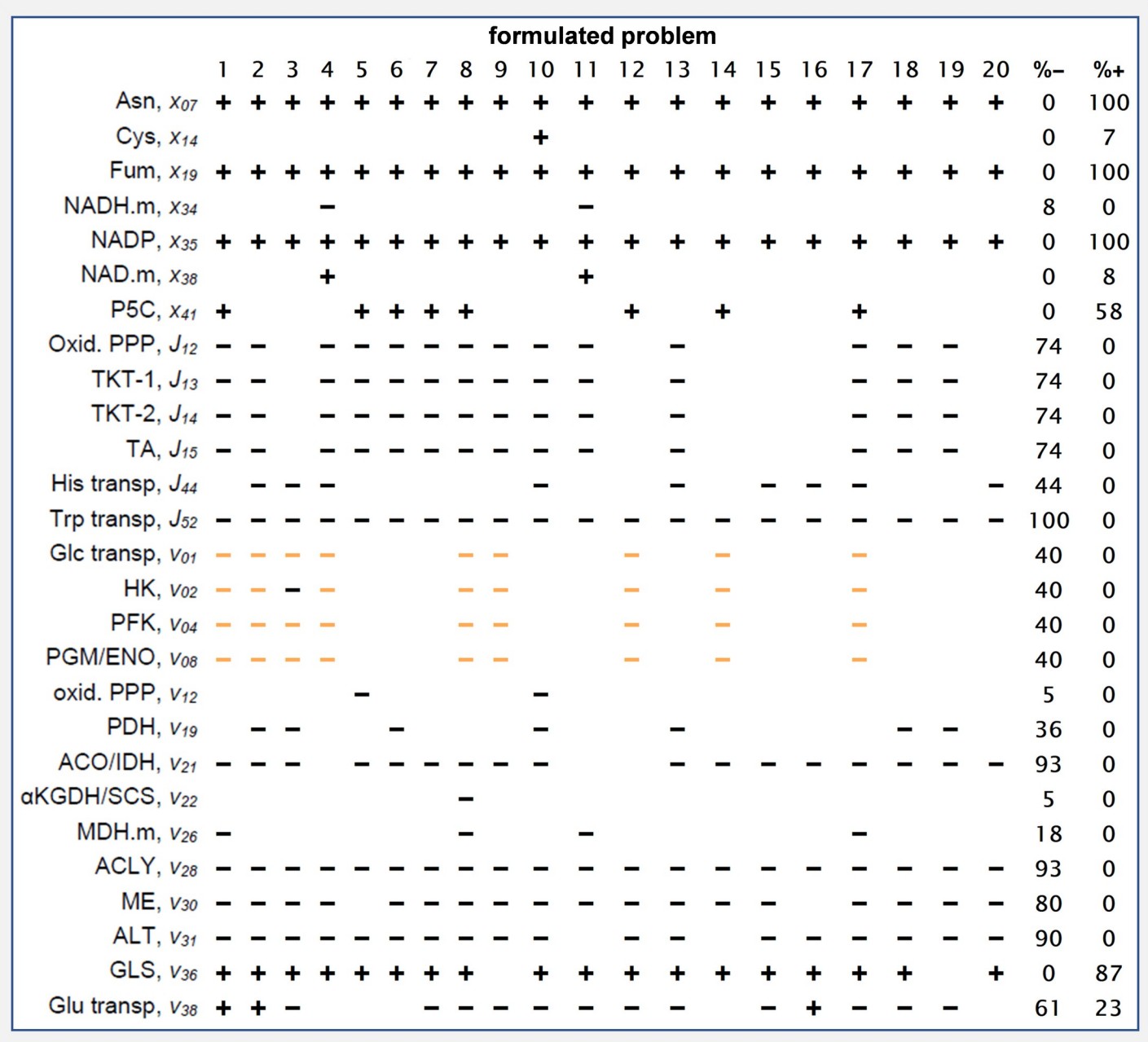

**Fig 4. Identification of fixed signs. Cancer-case study.** Each column with a number in the top is equivalent to the last column of the table in the panel C (sign) for the glycolysis-case study in Fig 2, for each of the first 20 solved problems of the ensemble of 100 formulated problems. Percentages of negative (% -) and positive (% +) signs refer to the solved problems. The average % gain for all variables and the 100 formulated problems was 29%. In orange, signs dependent on the constraint in Eq (14) that assumes a coordinated regulation changing in parallel the individual activities of Glc transp, HK, PFK, and ENO. See Fig 5 for a numerical summary of the final domains. See Material and Methods for a supplementary description of the network and abbreviations.

enough to require a decrease in the individual activity of HK, as shown in the 3rd model formulation (Fig 4), while for the rest of the model formulations the additional constraints in Eq (14) connecting the changes in the activity of HK with those for Glc transp, PFK and ENO are needed.

| | initial domain | union of final domains | % gain | intersection of final domains | % gain |
|---|---|---|---|---|---|
| Asn, $x_{07}$ | $-3 < \Delta \log x_{07} < 3$ | $0.53 < \Delta \log x_{07} < 1.48$ | 84.2 | $0.63 < \Delta \log x_{07} < 1.06$ | 92.8 |
| Cys, $x_{14}$ | $-3 < \Delta \log x_{14} < 3$ | $-0.63 < \Delta \log x_{14} < 1.28$ | 68.2 | $0.02 < \Delta \log x_{14} < 0.25$ | 96.2 |
| Fum, $x_{19}$ | $-3 < \Delta \log x_{19} < 3$ | $0.22 < \Delta \log x_{19} < 1.10$ | 85.3 | $0.27 < \Delta \log x_{19} < 0.97$ | 88.4 |
| NADH.m, $x_{34}$ | $-3 < \Delta \log x_{34} < 3$ | $-0.86 < \Delta \log x_{34} < 0.71$ | 73.8 | $-0.53 < \Delta \log x_{34} < -0.06$ | 92.1 |
| NADP, $x_{35}$ | $-3 < \Delta \log x_{35} < 3$ | $0.23 < \Delta \log x_{35} < 0.61$ | 93.7 | $0.23 < \Delta \log x_{35} < 0.23$ | 100 |
| NAD.m, $x_{38}$ | $-3 < \Delta \log x_{38} < 3$ | $-0.12 < \Delta \log x_{38} < 0.14$ | 95.6 | $0.01 < \Delta \log x_{38} < 0.09$ | 98.7 |
| P5C, $x_{41}$ | $-3 < \Delta \log x_{41} < 3$ | $-0.15 < \Delta \log x_{41} < 1.22$ | 77.2 | $0.22 < \Delta \log x_{41} < 0.32$ | 98.3 |
| Oxid. PPP, $J_{12}$ | $-0.66 < \Delta \log J_{12} < 0.12$ | $-0.66 < \Delta \log J_{12} < 0.12$ | 1.0 | $-0.66 < \Delta \log J_{12} < -0.28$ | 51.2 |
| TKT-1, $J_{13}$ | $-0.66 < \Delta \log J_{13} < 0.12$ | $-0.66 < \Delta \log J_{13} < 0.11$ | 1.2 | $-0.66 < \Delta \log J_{13} < -0.28$ | 51.3 |
| TKT-2, $J_{14}$ | $-0.66 < \Delta \log J_{14} < 0.12$ | $-0.66 < \Delta \log J_{14} < 0.11$ | 1.2 | $-0.66 < \Delta \log J_{14} < -0.28$ | 51.3 |
| TA, $J_{15}$ | $-0.66 < \Delta \log J_{15} < 0.12$ | $-0.66 < \Delta \log J_{15} < 0.11$ | 1.2 | $-0.66 < \Delta \log J_{15} < -0.28$ | 51.3 |
| His transp, $J_{44}$ | $-0.58 < \Delta \log J_{44} < 0.05$ | $-0.58 < \Delta \log J_{44} < 0.03$ | 4.3 | $-0.58 < \Delta \log J_{44} < -0.06$ | 17.6 |
| Trp transp, $J_{52}$ | $-0.64 < \Delta \log J_{52} < 0.93$ | $-0.48 < \Delta \log J_{52} < -0.16$ | 79.4 | $-0.48 < \Delta \log J_{52} < -0.16$ | 79.4 |
| Glc transp, $v_{01}$ | $-3 < \Delta \log v_{01} < 3$ | $-1.31 < \Delta \log v_{01} < 0.85$ | 64.0 | $-0.31 < \Delta \log v_{01} < -0.14$ | 97.2 |
| HK, $v_{02}$ | $-3 < \Delta \log v_{02} < 3$ | $-1.31 < \Delta \log v_{02} < 0.85$ | 64.0 | $-0.31 < \Delta \log v_{02} < -0.14$ | 97.2 |
| PFK, $v_{04}$ | $-3 < \Delta \log v_{04} < 3$ | $-1.31 < \Delta \log v_{04} < 0.85$ | 64.0 | $-0.31 < \Delta \log v_{04} < -0.14$ | 97.2 |
| PGM/ENO, $v_{08}$ | $-3 < \Delta \log v_{08} < 3$ | $-1.31 < \Delta \log v_{08} < 0.85$ | 64.0 | $-0.31 < \Delta \log v_{08} < -0.14$ | 97.2 |
| oxid. PPP, $v_{12}$ | $-3 < \Delta \log v_{12} < 3$ | $-3 < \Delta \log v_{12} < 2.14$ | 14.3 | $-1.22 < \Delta \log v_{12} < -0.11$ | 81.4 |
| PDH, $v_{19}$ | $-3 < \Delta \log v_{19} < 3$ | $-3 < \Delta \log v_{19} < 1.88$ | 18.7 | $-1.42 < \Delta \log v_{19} < -0.49$ | 84.5 |
| ACO/IDH, $v_{21}$ | $-3 < \Delta \log v_{21} < 3$ | $-0.79 < \Delta \log v_{21} < 0.06$ | 85.8 | no intersection | n/a |
| αKGDH/SCS, $v_{22}$ | $-3 < \Delta \log v_{22} < 3$ | $-3 < \Delta \log v_{22} < 2.8$ | 3.3 | $-0.57 < \Delta \log v_{22} < -0.05$ | 91.2 |
| MDH.m, $v_{26}$ | $-3 < \Delta \log v_{26} < 3$ | $-3 < \Delta \log v_{26} < 1.85$ | 19.2 | $-1.53 < \Delta \log v_{26} < -0.85$ | 88.6 |
| ACLY, $v_{28}$ | $-3 < \Delta \log v_{28} < 3$ | $-3 < \Delta \log v_{28} < -0.25$ | 54.2 | $-3 < \Delta \log v_{28} < -2.28$ | 88.0 |
| ME, $v_{30}$ | $-3 < \Delta \log v_{30} < 3$ | $-1.23 < \Delta \log v_{30} < 0.39$ | 73.0 | no intersection | n/a |
| ALT, $v_{31}$ | $-3 < \Delta \log v_{31} < 3$ | $-1.4 < \Delta \log v_{31} < 0.08$ | 75.3 | $-0.8 < \Delta \log v_{31} < -0.47$ | 94.4 |
| GLS, $v_{36}$ | $-3 < \Delta \log v_{36} < 3$ | $-0.05 < \Delta \log v_{36} < 1.18$ | 79.6 | $0.39 < \Delta \log v_{36} < 0.65$ | 95.5 |
| Glu transp, $v_{38}$ | $-3 < \Delta \log v_{38} < 3$ | $-0.26 < \Delta \log v_{38} < 0.16$ | 93.1 | no intersection | n/a |

**Fig 5. A numerical summary of the final domains. Cancer-case study.** Interval unions and interval intersections of the ensemble of final domains for each variable in Fig 4. All logarithms are to base two.

For those variables that were more occasionally associated with repeated negative or positive signs or associated with a mixture of signs, such as glutamate (Glu) transport (Glu-transp), the presence of signs was more dependent on the uncertainty associated with the sampling of metabolite elasticities. Although an important level of uncertainty was permitted, the assumptions related to levels of saturation, moiety conservations, inhibitions, and activations, altogether affected the numerical value of the bounds of final domains in almost all variables. This can be shown in Fig 5 by comparing the interval unions and interval intersections of the final domains for each variable. Further availability of mechanistic data should constrain the

metabolic elasticities, where the maximum restriction is achieved with a complete kinetic model, as for the glycolysis-case study.

## Supporting the role as metabolic drivers of differentially expressed genes

In our previous work [50], we demonstrated that the inhibition of CDK4 and CDK6 resulted in the perturbation of fundamental regulators of the metabolic activity. Thus, in response to CDK4/6 inhibition, in combination with MYC's upregulation and the activation of the PI3K/Akt-mTOR signaling axis, the hypoxia-inducible factor 1 (HIF1) was strongly downregulated. A detailed screening was performed among the differentially expressed genes detected in an Affymetrix GeneChip Human Genome U133 Plus 2.0 Array. A subset of them was experimentally verified by qRT-PCR (Fig 3). Also, our experimental observations supported previous reports sustaining that part of these differentially expressed genes were HIF1-dependent genes [69–71]. The effect on gene expression of hypoxia and CDK4/6 inhibition was the opposite for several genes [50], therefore suggesting that the effects of CDK4/6 inhibition were in part driven by HIF1 downregulation. In particular, among the selected genes, *SLC2A3*, *HK2*, *PFKFB4*, *ENO2*, *PDK1*, and *PDK3* were upregulated in response to hypoxia and downregulated in response to CDK4/6 inhibition. Based on these evidences we assumed a coordinated regulation by the transcriptional regulator HIF1 of glycolytic *SLC2A3*, *HK2*, *PFKFB4*, and *ENO2* genes, and therefore the coordinated regulation of the individual activities of their encoded products (Glc transp, HK, PFK, and ENO, respectively) in Eq (14).

For illustrative purposes, those genes coding for products known to directly affect individual activities of the core part of the model are listed in Table 2. The table provides the percentages of identified changes in the activities affected by their encoded products.

The percentages of negative and positive signs were used to assess if the computational analysis qualitatively supports a possible role as molecular drivers of the metabolic adaptations for the measured changes in gene expression. There was, in general, a good correspondence that supported this role for most of these analyzed genes. In Table 2 (last column), we have classified these changes as "supported" or "supported, but sampling dependent" and "not supported, but sampling dependent". The role as key drivers of the metabolic adaptation for *IDH2* and *GLS1* was supported by the high percentages of signs indicating that the changes predicted in metabolic activities are going in the same direction as the changes expected from the

**Table 2. Decreases/increases expected from measured genes differentially expressed and decreases/increases predicted in the metabolic activities affected by their encoded products.**

| Gene ID | qRT-PCR | Affy.GCh | Affected ind. activity ID | Expected | Predicted | | Role as a driver of the metabolic adapt. |
|---------|---------|----------|---------------------------|----------|-----------|-----------|------------------------------------------|
| | | | | | % - | % + | |
| *SLC2A3* | -0.8 | -1.2 | Glc-transp/HK/PFK/PGM/ENO(HIF1) | – | 40 | 0 | supported, but sampling dependent |
| *HK2* | -0.9 | -0.7 | | | | | |
| *PFKFB4* | -1.8 | -0.8 | | | | | |
| *ENO2* | -2.3 | -1.3 | | | | | |
| *PDK1* | -1.2 | -0.5 | PDH | + | 36 | 0 | not supported, but sampling dependent |
| *PDK3* | -1.2 | -1.3 | | | | | |
| *IDH2* | -0.7 | -0.6 | ACO/IDH | – | 93 | 0 | supported |
| *GLS1* | +1.0 | +0.6 | GLS | + | 0 | 87 | supported |

Numbers for qRT-PCR and Affymetrix GeneChips (Affy-GCh) are $\log_2$-fold changes. Expected signs in the changes for individual activities are based on the direction observed in gene expression, while the predicted percentages of negative and positive signs correspond to the percentages in Fig 4. See Table 3 for a description of the role of the encoded products of the selected genes.

**Table 3. Description of the role of the encoded products of the selected genes.**

| Gene ID | Name; direct metabolic role | Affected ind. activity ID |
|---------|-----------------------------|---------------------------|
| SLC2A3 | Solute carrier family 2, facilitated glucose transporter member 3; facilitative Glc transporter that mediates the uptake of Glc and various other monosaccharides across the cell membrane. | $R_{01}$ (Glc-transp) |
| HK2 | Hexokinase-2; catalyzes the initial step of glycolysis, the phosphorylation of Glc to produce G6P. Predominant isoform found in skeletal muscle. | $R_{02}$ (HK) |
| PFKFB4 (*) | 6-phosphofructo-2-kinase/fructose-2,6-bisphosphatase 4; bifunctional enzyme that catalyzes the synthesis (kinase activity) or degradation (phosphatase activity) of fructose 2,6-bisphosphate (F26BP), an allosteric regulator that activates the glycolytic PFK resulting in increased glycolysis. Isoform originally identified in testis, over-expressed in human cancers, functions predominantly to synthesize F26BP (has far more kinase activity than phosphatase activity), therefore increasing the glycolytic flux. | $R_{04}$ (PFK) |
| ENO2 | Gamma-enolase; catalyzes the dehydration of 2-phosphoglycerate to PEP as part of the glycolytic pathway. Isoform primarily expressed by mature neurons and cells of neuronal origin. | $R_{08}$ (PGM/ENO) |
| PDK1 PDK3 | Pyruvate dehydrogenase (acetyl-transferring) kinase isozymes 1/3, mitochondrial; inactivate by phosphorylation the pyruvate dehydrogenase complex (PDH) activity, which catalyzes the first step of the Krebs cycle. | $R_{19}$ (PDH) |
| IDH2 | Isocitrate dehydrogenase [NADP], mitochondrial isozyme; catalyzes the oxidative decarboxylation of isocitrate to α-ketoglutarate. | $R_{21}$ (ACO/IDH) |
| GLS1 | Kidney glutaminase (KGA) and glutaminase C (GAC) isoforms; catalyze the hydrolysis of Gln to Glu and ammonia. Alternative splicing isoforms ubiquitously expressed in various tissues. Overexpression of both isoforms confirmed by isoform-specific antibodies in our HCT116 cells. | $R_{36}$ (GLS) |

(*) [72]

measurements in gene expression. For the rest, the lower percentages do not allow for any conclusion, rather than this will depend on the sampling of metabolite elasticities.

Assuming the coordinated regulation of glycolytic *SLC2A3*, *HK2*, *PFKFB4*, and *ENO2* by the constraints in Eq (14), the required decreases in their affected activities followed the measured reduction in HIF1. This assumption strongly reduced the space of solutions, facilitating the appearance of signs supporting the requirement of such transcriptional co-regulation, as measured experimentally. However, the signs (in orange, Fig 4) disappeared when the constraints assuming the coordinated regulation were not included.

Although some negative signs were predicted for the HIF1-dependent mitochondrial pyruvate dehydrogenase complex (PDH) activity, the downregulation of *PDK1*/*PDK3* should drive an increase in PDH activity. This discrepancy is helpful as it indicates that additional information for the post-translationally regulated PDH activity is required to explain the observed behavior.

Finally, the required increase in the activity of GLS, supported by western blot analysis, followed the expected up-regulation of MYC-dependent *GLS1*, since MYC upregulates GLS activity by suppressing the expression of miR-23a and miR-23b, which target the *GLS1* transcripts [73]. Indeed, in our published analysis, the combined inhibition of GLS activity and CDK4/6 was experimentally validated as a promising synergetic combination for the efficient and selective killing of cancer cells.

## Discussion

Exploring the dependencies of metabolic variations measured in concentrations, fluxes, and transporter and enzyme activities can be done using kinetic models that accurately simulate the system behavior. Undeniably, coupling kinetic models with optimization methods

provides great advantages in metabolic modeling. However, even if these models are based on approximated rate laws, they require detailed knowledge of enzyme kinetics inaccessible *in vivo*. To overcome this limitation, sampling strategies can be applied in a context of partial knowledge, although with the disadvantage of having a large space of solutions to be explored. Coupled with sampling strategies, our proposed method exploits the advantages of both approaches. First, a limited ensemble of control-coefficient matrices is generated by sampling metabolite elasticities, which together describe the partial knowledge of the system behavior. This is then used to formulate problems based on linear constraints. Finally, the dependencies of the reaction fluxes, concentrations, and individual activities are exhaustively explored by linear optimization methods in light of these linear constraints.

With the goal of identifying molecular drivers of the changes observed in metabolic adaptations to perturbations, the applied analysis is based on a combination of the MCA description of response coefficients as a function of special elasticities and control coefficients [30–33], and the subsequent reformulation as part of a linear optimization-based strategy for bound contraction in the context of large changes. This reformulation takes advantage of linearization around a reference steady-state and therefore cannot be applied for large changes with a quantitative aim. However, it can be used to capture the trends of the changes, increases (positive) or decreases (negative). Accordingly, the objective was to obtain a collection of positive and negative signs, whose repetition will allow us to assess the significance of these changes when the problem is coupled with uncertainty. First, the glycolysis-case study allowed us to illustrate this method using a unique matrix of control coefficients derived from an unambiguously reconstructed system around a steady-state. Second, the study of the effects of CDK4/6 inhibition, coupled with uncertainty, was performed in more realistic conditions, i.e., for a more complex perturbation and with partial knowledge, solving an ensemble of problems derived from sampling techniques. The interpretation of the solutions for this ensemble of problems must be made from a more global perspective [65]. Although each solved problem satisfies the constraints associated with our knowledge of the metabolic system, the ensemble describes different possible behaviors as a measure of uncertainty. By analyzing the set of resulting final domains for each variable, the requirement for negative or positive signs in just a few cases is of null relevance. In contrast, repeated signs in the entire set are very relevant. For some variables, the required changes were largely independent of the sampling of metabolite elasticities and dependent on the applied observations and assumptions. The set of 100 ensemble formulated problems was sufficient to provide a qualitative assessment of the significance of the changes. However, this number should be adapted depending on the size of the analyzed problem, and a more robust test with statistical value could be addressed in future work.

In summary, in the context of the well-known central carbon metabolism, therefore appropriate to illustrate the proposed methodology, this analysis was applied to support the role as metabolic drivers of genes differentially expressed. Our procedure successfully enabled the inference of required changes, although not necessarily sufficient, to sustain the whole set of constraints and inequalities associated with the mixture of observations and assumptions used to characterize a metabolic adaptation.

## Material and methods

### Glycolysis-case study

**Background.**   This case study was mostly based on a published kinetic model covering the upper glycolysis [48], which was derived from experiments performed on mice muscle extracts. This published model describes a linear pathway at a given steady-state. In our glycolysis-case study, one ramification and one moiety conservation were added by including the

first step of the oxidative PPP, which was taken from another published kinetic model in rat liver cells [49]. For all reactions were available kinetic laws, parameters, as well as steady-state concentrations and fluxes.

**Abbreviations.** G6P, $x_1$, glucose-6-phosphate; F6P, $x_2$, fructose-6-phosphate; FbP, $x_3$, fructose-1,6-diphosphate; NADP (NADP$^+$) and NADPH, $x_4$ and $x_5$, oxidized and reduced forms of nicotinamide adenine dinucleotide phosphate; HK, $R_1$, hexokinase; GPI, $R_2$, glucose-phosphate isomerase; PFK, $R_3$, phosphofructokinase; ALD, $R_4$, aldolase; G6PD, $R_5$, glucose-6-phosphate dehydrogenase; and NADPase, $R_6$, irreversible process accounting for all processes oxidizing NADPH.

**Network description.** The network analyzed in this glycolysis-case study includes five system-dependent metabolites and six enzyme-catalyzed reactions, with one moiety conservation. Fig 1 provides a complete scheme of the associated kinetic model, including also the associated system of ODEs, rate laws for all the enzyme-catalyzed reactions, parameter values, and stoichiometric dependencies of fluxes and concentrations.

**Model description using control coefficients.** A unique set of control coefficients was derived from the detailed kinetic model, therefore permitting a unique problem formulation. Control coefficients were estimated by following this procedure: 1) steady-state concentrations and fluxes are calculated by setting the initial conditions and solving the system of ODEs; 2) metabolite elasticities are calculated using the steady-state concentrations and fluxes; 3) stoichiometric flux and concentration dependencies are derived and used in the form of flux and concentration ratios, together with metabolite elasticities, to calculate control coefficients by applying the matrix method developed by Cascante et al. [31,32] (panel D in Fig 1).

## Cancer-case study

**Background.** A second study spanning all central carbon metabolism was also analyzed, based on our previous work covering the effects of CDK4/6 inhibition in the HCT116 colon cancer cell line [50]. The study characterized the metabolic reprogramming, both at the systemic and molecular levels, in response to CDK4/6 inhibition. To this aim, at the molecular level, downregulations and upregulations for gene levels were measured using transcriptome microarrays and qRT-PCR. At the systemic level, differences on the levels for all fluxes and some metabolites (alanine, aspartate, citrate, Glu, Mal, NADPH, pyruvate, and α-ketoglutarate) were measured for control cells (before perturbation) and CDK4/6-inhibited cells. A stoichiometric model for the central carbon metabolism was constructed and solved by applying SIRM techniques to estimate all flux distributions throughout the metabolic network, including forward and reverse reaction rates when required. For this, direct extracellular measurements, such as oxygen consumption, and consumption and production rates for Glc, lactate, and all amino acids, as well as protein synthesis rates, were combined with $^{13}$C isotopologue (mass isotopomer) enrichments measured in several metabolites. Such enrichments emerge from the propagation of $^{13}$C from labeled Glc and Gln to metabolites through the network and are informative of the underlying flux distribution. The metabolites analyzed for label propagation included lactate and amino acids from incubation media, glycogen, ribose from RNA, palmitate, and several other internal metabolites.

**Model abbreviations.** Metabolites: AcoA.c/AcoA.m, $x_{01}/x_{02}$, cytosolic/mitochondrial acetyl-CoA; ADP/ATP, $x_{03}/x_{09}$, adenosine di/triphosphate; αKG, $x_{04}$, α-ketoglutarate; Ala, $x_{05}$, alanine; Asn, $x_{07}$, asparagine; Asp, $x_{08}$, aspartate; Cit, $x_{11}$, citrate; CoA.c/CoA.m, $x_{12}/x_{13}$, cytosolic/mitochondrial coenzyme A; Cys, $x_{14}$, cysteine; FBP, $x_{18}$, fructose 1,6-bisphosphate; Fum, $x_{19}$, fumarate; Gln, $x_{23}$, glutamine; Glu, $x_{24}$, glutamate; Ile, $x_{27}$, isoleucine; Leu, $x_{29}$, leucine; Mal, $x_{31}$, malate; Met, $x_{32}$, methionine; NADH.c/NADH.m, $x_{33}/x_{34}$, cytosolic/mitochondrial

reduced form of nicotinamide adenine dinucleotide; NADP/NADPH, $x_{35}$/$x_{36}$, oxidized/ reduced form of nicotinamide adenine dinucleotide phosphate; NAD.c/NAD.m, $x_{37}$/$x_{38}$, cytosolic/mitochondrial oxidized form of nicotinamide adenine dinucleotide; OAA.c/OAA.m, $x_{39}$/ $x_{40}$, cytoplasmic/mitochondrial oxaloacetate; P5C, $x_{41}$, Δ1-pyrroline-5-carboxylate; PEP, $x_{43}$, phosphoenol pyruvate; Phe, $x_{44}$, phenylalanine; Pro, $x_{45}$, proline; Pyr, $x_{46}$, pyruvate; Ser, $x_{48}$, serine; Suc, $x_{49}$, succinate; and Val, $x_{53}$, valine. Transport and reactions processes: Glc-transp, $R_{01}$, glucose transport; HK, $R_{02}$, hexokinase; PFK, $R_{04}$, phosphofructokinase; PGM/ENO, $R_{08}$, phosphoglycerate mutase / enolase pool; PK, $R_{09}$, pyruvate kinase; oxid. PPP, $R_{12}$, oxidative part of pentose-phosphate pathway; TKT, $R_{13}$ and $R_{14}$, transketolase; TA, $R_{15}$, transaldolase; PDH, $R_{19}$, pyruvate dehydrogenase complex; ACO/IDH, $R_{21}$, aconitase / isocitrate dehydrogenase pool; αKGDH/SCS, $R_{22}$, α-ketoglutarate dehydrogenase / succinyl-CoA synthetase pool; SDH/CII, $R_{23}$, succinate dehydrogenase / oxidative phosphorylation from complex II of respiratory chain; FH, $R_{25}$, fumarate hydratase; MDH.m, $R_{26}$, malate dehydrogenase (mitochondrial); PC, $R_{27}$, pyruvate carboxylase; ACLY, $R_{28}$, citrate lyase; ME, $R_{30}$, malic enzyme; ALT, $R_{31}$, alanine transaminase; ASP, $R_{33}$/$R_{34}$, mitochondrial/cytosolic aspartate transaminase; Gln-transp, $R_{35}$, glutamine transport; GLS, $R_{36}$, glutaminase; GDH, $R_{37}$, glutamate dehydrogenase; Glu-transp, $R_{38}$, glutamate transport; His-transp, $R_{44}$, histidine transport; Trp-transp, $R_{52}$, tryptophan transport; and PYCR, $R_{60}$, pyrroline-5-carboxylate reductase.

**Network description.** The network analyzed in this cancer-case study includes 53 system-dependent metabolites (S1 Table) and 76 transport processes and enzyme-catalyzed reactions (S2 Table), with moiety conservations involving six pairs of metabolites (ACoA.c/CoA.c; ACoA.m/CoA.m; ATP/ADP; NAD.c/NADH.c; NAD.m/NADH.m; and NADP / NADPH). The associated stoichiometric model did not include rate laws, but rather just the reaction stoichiometry. A subset of the transport and reaction processes ($R_{01}$ –$R_{38}$) conforms a core network, including all processes with a significant flux. This core network includes the enzyme-catalyzed reactions for glycolysis, glutaminolysis, PPP, and TCA cycle, together with the measured uptake of Glc, Gln, and Ser, the release of Ala and Glu, and the fueling of mitochondrial respiration by NADH and succinate for ATP production. Fig 3 provides a scheme of the core network. The rest of the reactions ($R_{39}$ –$R_{76}$) are associated with simplified pools of reactions accounting for boundary processes, such as the release, uptake, and oxidation of the rest of amino acids, fatty acid synthesis, and glycogen synthesis. Among these simplified processes are protein synthesis, which was included to balance the exchange and utilization of amino acids, reactions for ATP and NADPH utilization, and the recycling of mitochondrial acetyl-CoA. All of them were included to have appropriate balances of productions and consumptions.

**Model description using control coefficients.** In a context of uncertainty, multiple problem formulations were solved, each one associated with a complete set of control coefficients. To estimate each matrix of control coefficients, all metabolite elasticities were simultaneously generated by applying random sampling. The sampling of metabolite elasticities was done satisfying different constraints, including restricted domains for them.

A first restriction that can be applied refers to the "positive" role of substrates and activators and the "negative" role of products and inhibitors:

$\varepsilon_M^v > 0$; $M$ is a substrate or activator of $v$.

$\varepsilon_M^v < 0$; $M$ is a product or inhibitor of $v$.

where the lower or upper bound, respectively for positive or negative elasticities, is set to a value of zero and corresponded to a situation of total saturation by $M$. However, the values of the metabolite elasticities also depend on other constraints. For a particular transport [74,75]

or enzyme-catalyzed reaction [15], characteristics such as the particular mechanism, the level of saturation, and the displacement of the reaction with respect to the chemical equilibrium determine the domains of possible values of the metabolite elasticities for this process [27,66–68,76–78]. Thus, taking a general reversible reaction:

$$v = vf - vr \tag{15}$$

where $v$ corresponds to the net rate of the reversible reaction, $vf$ the forward reaction rate and $vr$ the reverse reaction rate, the values of metabolite elasticities for $vf$ and $vr$ can be additionally restricted. The elasticity for a mass-action rate law is equal to the order of reaction of the reactants, while the elasticities derived for more complex rate laws, such as those associated with mechanisms for mediated transport and enzyme-catalyzed reactions, can range from zero to different values depending on the mechanism and the curve of saturation. For example, for reactions following cooperativity, which follow a non-hyperbolic curve of saturation, the limit is the Hill coefficient ($0 < \varepsilon_S^{vf} < h$), such as for the reaction catalyzed by PFK in the glycolysis-case model (Fig 1). For transport and enzyme-catalyzed reactions following a hyperbolic Michaelis-Menten type curve of saturation, all metabolite elasticities of the forward and reverse reaction rates range between zero and one [27,66,68]:

$$0 < (\varepsilon_S^{vf}, \varepsilon_P^{vr}) < 1; -1 < (\varepsilon_P^{vf}, \varepsilon_S^{vr}) < 0 \tag{16}$$

where 0 means total saturation and $S$ and $P$ refer to a substrate and a product, respectively, in reactions with one or more substrates and products. The elasticities for the overall reaction rate $v$ will be a function of these elasticities for the forward and reverse reaction rates and the levels of displacement with respect to the equilibrium.

Assuming, for simplicity, that all modeled transport and reaction-steps follow a Michaelis-Menten type saturation, with metabolite elasticities for all forward and backward reaction rates ranging between 0 and +1 or -1 (Eq (16)), the limits for the elasticities of the net reaction rates can be described as:

$$0 < \varepsilon_S^v < N; -N < \varepsilon_P^v < 0 \tag{17}$$

where $N$ depends on the level of displacement from equilibrium. At steady-state, the net, forward and reverse reaction rates can be expressed as a function of the disequilibrium ratio ($\rho$) [16],

$$vf = \frac{v}{1 - \rho} \geq 0, vr = \frac{\rho \times v}{1 - \rho} \geq 0 \tag{18}$$

where,

$$0 < \rho = \frac{vr}{vf} = \frac{\Gamma}{K_{eq}} < \infty, \ \Gamma = \frac{products}{substrates}, \ \text{and at equilibrium } \Gamma = K_{eq} \tag{19}$$

and the following equation can describe the metabolite elasticity of the net rate:

$$\varepsilon_R^v = \frac{vf \times \varepsilon_R^{vf} - vr \times \varepsilon_R^{vr}}{v} = \frac{\varepsilon_R^{vf} - \rho \times \varepsilon_R^{vr}}{1 - \rho} \tag{20}$$

where $v$ can be positive or negative, $R$ is a reactant (substrate or product), in reactions with one or more substrates and products, $K_{eq}$ is the equilibrium constant, $\Gamma$ is the mass-action ratio, and $\rho$ is the disequilibrium ratio. Given three extreme situations:

- $\rho = 0$, the reaction is irreversible ($v = vf$ and $vr = 0$) and $\varepsilon_R^v = \varepsilon_R^{vf}$

- $\rho = 1$, the reaction is in equilibrium ($vf = vr$) and $\varepsilon_R^v = \infty, -\infty$ (indeterminate)

- $\rho = \infty$, the reaction is irreversible ($vf = 0$ and $v = -vr$) and $\varepsilon_R^v = \varepsilon_R^{vr}$

Accordingly, $N$ in (17) will tend towards infinity for metabolite concentrations close to equilibrium [15,79,80].

Furthermore, regarding metabolite elasticities for both forward and backward directions, they are not independent, and the following constraints should also be considered [27,68]:

$$\varepsilon_R^{vf} - \varepsilon_R^{vr} = 1; \; R = S \text{ for } 0 < \rho < 1; \; R = P \text{ for } 1 < \rho < \infty \tag{21}$$

$$\varepsilon_R^{vf} - \varepsilon_R^{vr} = -1; \; R = P \text{ for } 0 < \rho < 1; \; R = S \text{ for } 1 < \rho < \infty \tag{22}$$

Taking $0<\rho<1$, and therefore for $v>0$, by substituting in Eq (20) $\varepsilon_S^{vr}$ and $\varepsilon_S^{vf}$ according to the equality in Eq (21) for S ($\varepsilon_R^{vr} = \varepsilon_R^{vf} - 1$ and $\varepsilon_R^{vf} = \varepsilon_R^{vr} + 1$) and $\varepsilon_P^{vr}$ and $\varepsilon_P^{vf}$ according to the equality in Eq (22) for P ($\varepsilon_R^{vr} = \varepsilon_R^{vf} + 1$ and $\varepsilon_R^{vf} = \varepsilon_R^{vr} - 1$), the following equations are derived [15,79], respectively:

$$\varepsilon_S^v = \frac{\rho}{1-\rho} + \varepsilon_S^{vf} = \frac{1}{1-\rho} + \varepsilon_S^{vr} \tag{23}$$

$$\varepsilon_P^v = -\frac{\rho}{1-\rho} + \varepsilon_P^{vf} = -\frac{1}{1-\rho} + \varepsilon_P^{vr} \tag{24}$$

which permitted in our calculations the indirect calculation of $\varepsilon_S^v$ and $\varepsilon_P^v$ from the sampled values for $\varepsilon_S^{vf}$ and $\varepsilon_P^{vf}$, respectively. The first and second right-hand terms in Eqs (23) and (24) correspond to the so-called regulatory saturation term and thermodynamic or mass action term, respectively [79,80]. It should be noted that the application of Eq (24) implies that even for reactions far from equilibrium ($\rho = 0$), we will account for products' effect, consistent with the fact that in multienzyme systems the products' concentrations are rarely zero [15,81].

Accordingly, for each problem formulation, metabolite elasticities with respect to substrates and products were sampled for the forward flux (Eq (16)), and then used to calculate the elasticities for the net reactions applying Eq (23) for substrates and Eq (24) for products. This required an estimation of the disequilibrium ratios (S2 Table). Most of the reactions associated with transport and simplified pools of reactions accounting for boundary processes were assumed to be far from equilibrium. For some other processes, disequilibrium ratios known to be closer to equilibrium in biological conditions were set to $\rho = 0.9$, except for some cases for which $\rho$ were corrected to lower values using as an approximation SIRM-estimated reverse to forward rates (Eq (19)). The upper level of 0.9 for $\rho$ was selected as a compromise to avoid unrealistic values of control coefficients. In addition, although kinetic details were unknown, some constraints and bounded domains for metabolite elasticities were added from literature. These included constraints associated with known substrate/product competitive inhibitions [82–85]: ATP/ADP for PK, PC and ACLY; PEP/Pyr for PK; CoA.m/ACoA,m for PDH; Suc/Fum for SDH; αKG/Pyr and Glu/Ala for ALT; αKG/OAA and Glu/Asp for AST; Glu/αKG for GDH; and P5C/Pro and NADP/NADPH for PYCR. Then, an additional constraint was considered that must be satisfied for each of these competitive inhibitions involving a substrate and a product [76]:

$$0 < \varepsilon_S^v + \varepsilon_P^v = \varepsilon_S^{vf} + \varepsilon_P^{vf} = \varepsilon_S^{vr} + \varepsilon_P^{vr} < 1 \tag{25}$$

Also, elasticities accounting for other inhibitions and activations were considered, which are affecting the activity of the following enzymes: PFK inhibited by ATP and Cit, and activated by ADP [86]; PK inhibited by Ala, Cys, Met, Phe, Val, Leu, Ile and Pro, and activated by FBP and Ser [87]; PC inhibited by Glu [88]; PC activated by ACoA.m [89]; SDH inhibited by Mal [90]; SDH inhibited by OAA.m [91]; GLS activated by ADP [92]; and GDH inhibited by ATP (GTP), and activated by ADP and Leu [93]. For them, values were sampled into ranges between zero and one:

$$0 < \varepsilon_A^v < 1; -1 < \varepsilon_I^v < 0 \tag{26}$$

where $A$ and $I$ refer to an activator and an inhibitor, respectively. These constraints and bounded domains were imposed during the sampling of metabolite elasticities.

Following the applied matrix formulation [31,32], control coefficients are not only a function of metabolite elasticities, but also a function of the ratios between dependent fluxes and concentrations. The steady-state values for the flux ratios were approximated using average values for the reaction fluxes for control cells (before perturbation) and for CDK4/6 inhibited cells (S2 Table), which presented close flux distributions. For the moiety conservations involving six pairs of metabolites, the ratios for concentrations were set from literature as: 1) ACoA. c/CoA.c = 0.019 (cytosolic acetyl-CoA and CoA) and ACoA.m/CoA.m = 0.33 (mitochondrial acetyl-CoA and CoA) [94]; 2) ATP/ADP = 8, NAD.c/NADH.c = 120 and NAD.m/NADH. m = 6 [95]; and 3) NADP/NADPH = 0.52 [50].

Only matrices with all control coefficients within a -3.5 to +3.5 range were selected. As for the differences in concentrations, fluxes, and individual activities, values outside this enclosure were not accepted. Although it is arbitrarily set, this enclosure was added to avoid models with unrealistic sensitivities. In our particular application, to select 100 matrices of control coefficients, a total of 4686 were generated.

The S2 Table contains the list of reactions, stoichiometry, reaction fluxes, and disequilibrium ratios.

**Initial domains.** The initial domains for differences in all the reaction fluxes ($lb_{J_j}^{in} \leq \Delta \log J_j \leq ub_{J_j}^{in}$) and some of the metabolite concentrations ($lb_{x_i}^{in} \leq \Delta \log x_i \leq ub_{x_i}^{in}$) were restricted according to experimental observations comparing measurements at the states before the perturbation ($\log J_{j_o}$, $\log x_{i_o}$) and after the perturbation ($\log J_{j_p}$, $\log x_{i_p}$).:

$$\Delta \log x_i = \log x_{i_p} - \log x_{i_o} + \sigma; \Delta \log J_j = \log J_{i_p} - \log J_{j_o} + \sigma \tag{27}$$

A logarithm base of two was used, and therefore the differences are expressed as log$_2$-fold changes. Lower and upper bounds were estimated from the lower and upper bounds of confidence intervals (fluxes) and mean ± standard deviation (concentrations) of the measured values after perturbation. Other factors, such as variations in volume, could lead to a proportional bias of all fluxes, concentrations, and activities in one state with respect to the other. In the original study, quantities for fluxes and concentrations were expressed as quantities *per* cell. Assuming a uniform distribution of the metabolic species in the cells, a normalization or correcting factor $\sigma$ was applied to compare quantities per cell volume. The S3 Table contains the list of initial domains with additionally reduced bounds based on the measured differences.

## Calculations

All calculations were done using "*Wolfram Mathematica 11/12*" (www.wolfram.com). In particular, the function "*LinearProgramming*" was used to apply LP.

A Mathematica notebook, *"DomainSolver.nb"*, is provided to solve the final domains and identify negative and positive signs starting from calculated control coefficients and initial values. This makes the procedure fully reproducible, permitting the generation of the final domains used in Fig 2 (glycolysis-case) and Figs 4 and 5 (cancer-case). The Mathematica notebook contains an interactive script that requires to open two (glycolysis-case) or three (cancer-case) excel files (xlsx) with: 1) one matrix (glycolysis-case) or an ensemble of control-coefficient matrices (cancer-case); 2) list of initial domains (glycolysis-case and cancer-case); and 3) additional constraints for individual activities (only cancer-case, Eqs (13) and (14)). Also, a second Mathematica notebook, *"ControlSolver.nb"*, is provided to generate the control coefficients by random sampling of metabolite elasticities for the cancer-case study. This notebook contains another interactive script that requires to open one (cancer-case) excel file (xlsx) with a description of the network structure, flux values, disequilibrium ratios, substrate/product competitive inhibitions (Substrate competitions), inhibitors, activators, and moiety conservations. Two documents are provided with the instructions for use. These files are freely available on Zenodo at link http://dx.doi.org/10.5281/zenodo.5081161.

## Supporting information

**S1 Table. List of metabolites.**
(PDF)

**S2 Table. List of reactions, stoichiometry, reaction fluxes, and disequilibrium ratios.**
(PDF)

**S3 Table. List of initial domains with reduced bounds.**
(PDF)

## Author Contributions

**Conceptualization:** Pedro de Atauri, Marta Cascante.

**Data curation:** Pedro de Atauri, Míriam Tarrado-Castellarnau, Josep Tarragó-Celada, Marta Cascante.

**Formal analysis:** Pedro de Atauri, Míriam Tarrado-Castellarnau, Josep Tarragó-Celada, Josep Joan Centelles, Marta Cascante.

**Funding acquisition:** Marta Cascante.

**Investigation:** Pedro de Atauri, Míriam Tarrado-Castellarnau, Josep Tarragó-Celada, Marta Cascante.

**Methodology:** Pedro de Atauri, Carles Foguet, Effrosyni Karakitsou, Marta Cascante.

**Project administration:** Pedro de Atauri, Míriam Tarrado-Castellarnau, Josep Joan Centelles, Marta Cascante.

**Resources:** Josep Joan Centelles, Marta Cascante.

**Software:** Pedro de Atauri.

**Supervision:** Pedro de Atauri, Míriam Tarrado-Castellarnau, Josep Joan Centelles, Marta Cascante.

**Validation:** Pedro de Atauri, Míriam Tarrado-Castellarnau, Josep Tarragó-Celada, Marta Cascante.

**Visualization:** Pedro de Atauri.

**Writing – original draft:** Pedro de Atauri, Míriam Tarrado-Castellarnau, Marta Cascante.

**Writing – review & editing:** Pedro de Atauri, Míriam Tarrado-Castellarnau, Josep Tarragó-Celada, Carles Foguet, Effrosyni Karakitsou, Josep Joan Centelles, Marta Cascante.

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
