## [Decision Letter · Decision Letter 0]

24 Feb 2021

Dear Dr. Cascante,

Thank you very much for submitting your manuscript "Integrating systemic and molecular levels to infer key drivers sustaining metabolic adaptations" for consideration at PLOS Computational Biology.

As with all papers reviewed by the journal, your manuscript was reviewed by members of the editorial board and by several independent reviewers. In light of the reviews (below this email), we would like to invite the resubmission of a significantly-revised version that takes into account the reviewers' comments.

Please give particular attention to the data availability, and providing more accurate presentation on novelty and general applicability.

We cannot make any decision about publication until we have seen the revised manuscript and your response to the reviewers' comments. Your revised manuscript is also likely to be sent to reviewers for further evaluation.

Sincerely,

Kiran Raosaheb Patil, Ph.D.

Associate Editor

PLOS Computational Biology

Feilim Mac Gabhann

Editor-in-Chief

PLOS Computational Biology

Please give particular attention to the data availability, and providing more accurate presentation on novelty and general applicability.

Reviewer's Responses to Questions

**Comments to the Authors:**

Reviewer #1: Overview:

This manuscript presents a method to identify metabolites and enzyme-catalyzed reactions that significantly influence the metabolic response (reprogramming) to perturbations. The approach is applied for a simple model with detailed rate expressions, as well as a larger model. The method put forth in this paper is useful and relevant. However, details are missing the prevent a full understanding of the approach and its applicability.

Specific comments:

(1) Details regarding implementation of the approach are missing: how were the bounds of the control coefficients for sampling set? Is a set of 100 ensemble formulated problems sufficient? How are the fluxes determined for the larger model?

(2) For some of the predicted metabolic activities shown in Table 2 and compared to experimental data, less than the majority of the set of 100 solutions have only positive (or negative) fluxes. Can conclusions truly be drawn from this, without a consensus? Do the authors expect that a certain percentage is needed to make a clear conclusion?

(3) The paper describes one contradiction between the predicted metabolic activities and experimental data (PDH). Were there other discrepancies between the model and data? These should be described for full disclosure, as it helps the reader place the predictions and the approach in context.

(4) What does the last column in Table 2 mean?

(5) It is not clear how some constraints were established. For example, in the colon cancer model example, an assumption about coordinated regulation of a block of enzymes was made. How was this determined? What would the results be if that assumption was relaxed – would the results compare to experiments? Other places where additional constraints are applied but not explained: lines 305-307, red font in Fig. 2, Table S4.

(6) Is the black “-“ sign for the HK reaction in the 3rd model formulation a typo? Should this be orange/red?

(7) The authors have not explained what the union and intersection for the final domains mean.

(8) In the current form, it is just not clear how one could implement the approach, especially considering point 5 above. The authors can clarify how to apply their method.

Reviewer #2: The authors present a mathematical framework that combines metabolic control analysis with linear programming to estimate ranges of response coefficients in kinetic models with incomplete information. Although the work looks interesting, I find the structure of the manuscript is very hard to follow and I could not fully understand what are the advantages of the proposed method and how they differ from previous work. Please find some more detailed comments bellow.

- The results section begins by introducing basic definitions of metabolic control analysis (control coefficients, elasticities, summation and connectivity theorems). I don't understand why this is all introduced in the results section.

- The authors then show how the basic theorems of MCA can be re-written in a different structure that creates a linear system of equations. Throughout this section the authors often mention how different MCA and BST systems (GMA, S-Systems, lin-log) have different advantages. Again, I don't see why this is included in the results and makes it hard to understand which pieces of what is being presented is novel work and what is already known.

- The toy case study did not help to understand the method. For instance, Figure 1 shows that the control coefficients are derived from the ODE model, but does not explain how.

- When presenting the larger case study, called "Response to a large perturbation", the authors reformulate the previously introduced linear system, with a slightly different one that replaces the response coefficient with logarithms, under the justification that the one used in the first case study is only valid for infinitesimal perturbations, which not appropriate for real case studies. So why not present this formulation from the beginning?

- The final results of the cancer case study do seem relevant, and they are supported by the authors own experimental work, but it is very hard to access their relevance more clearly without a better understanding of the mathematical framework.

Reviewer #3: The manuscript describes a computational methodology based on combining metabolic control analysis with liner programming to identify major metabolic adaptions to small and large perturbations, such as those induced by drug therapy. As such, the work is thematically relevant to the scope of the journal. The overall idea is worthy of publication, however there are a number of concerns and questions that need to be addressed in order for the manuscript to be publishable.

A major concern comes from the authors’ claim that the methodology presented can address system responses to large perturbations. However, the whole premise of the work is based on linearization around some state(s), which by default is valid for small perturbations around those states. The validity of linearization of a very complex system when applying large perturbation is at the least questionable and needs to be properly justified.

Another major concern has to do with the novelty of the work presented. Performing linear optimization for metabolic control analysis (MCA) is certainly not novel, as also evidenced by the various references cited in the manuscript. Discussion of the novelty of the work, therefore needs to be significantly improved.

Also the mathematical discussion needs to be tightened up and sorted out in places.

Detailed points and questions are listed below to aid the authors in preparing an improved manuscript:

1. In the beginning of the results section (lines 150-152, the authors state that in this manuscript “MCA has been defined using slightly different formulations, which are also related with tis used in BST”. This phrase in addition to being syntactically wrong does not clarify the novelty of the work. In fact a large portion of the results section intermittently combines background information with seemingly new information, to the point that any potential novelty is hidden. I would recommend a rewrite, where the background is clearly discernible from the actual results.

2. Table 1 give definitions of sensitivity coefficients as used in the manuscript. A. It is not clear why dx_i/dNu_k is not a partial differential. In the same way dNu_k/dp should NOT be a partial differential. B. It is not clear which of these deifinitions are new as in one or another form they have been used before. This needs to be clarified as it it forms the basis of the work presented.

3. Line 184: It is clearly the chain rule and not chain’s rule applied here.

4. Equations 3 and 4 are derived by applying the chain rule on equations 1 and 2: A. The novelty (or not) of this formulation needs to be better discussed as seemingly references 31-34 us ethe same premise. B. It is not clear how these equations imply some kind of two level hierarchy. Response coefficients are direct functions of control coefficients, but once these are somehow calculated or estimated, then response coefficients are also known (fully or partially), so the two-level approach implied needs to be better thought or explained.

5. The “inference by bound contraction section is written pretty much as a literature review section and not particularly as a results one, so this needs to be seriously addressed.

6. Line 233-234: What are control coefficients fixed to? This is normally the most difficult information to obtain for a network

7. Line 248: In which cases is the problem not solvable? How is this assessed and/or addressed?

8. The authors are essentially “inverting” flux variability analysis procedure by fixing fluxes and calculating the min-max of response coefficients. This is a nice approach. It needs however to be clarified what is the final domain reduced from (as stated in line 258-259), i.e. what is the initial domain?

9. What is the reduced domain for the response coefficients in line 305-307?

10. Regarding the response to large perturbations section: The main question is how are equations 7-12 applicable as they are essentially linearizations? What makes this a valid approach for a complex highly nonlinear system? Linearisation for small perturbations is a logical approach. Applying this to large perturbations is a logical leap.

11. Taking into account the complexity of the second case study (as the glycolysis one is essentially used as a validation problem) how is it expected to know flux control coefficients and in addition those to be constants? In my view this is the trickiest information to obtain. If these are obtained by flux control analysis using constant response coefficients, how is this not undermining the information obtained in this work?

12. Line 370: Why is the initial domain set between -3 and 3? Is it just heuristics?

13. Line 437: how was reversibility (or not) decided and fixed for each of the reactions involved? This needs to be clarified in the manuscript.

14. In line 481, which is the original paper mentioned?

15. The “discussion” section explains a number of things that remain vague in the results section. So as mentioned above a significant rewrite need to take place.

16. Lin 731. It is not clear to what equations 20-22 are integrated. Which are the limits etc. This needs to be clearly discussed in the manuscript.

**Have all data underlying the figures and results presented in the manuscript been provided?**

Reviewer #1: Yes

Reviewer #2: None

Reviewer #3: None

PLOS authors have the option to publish the peer review history of their article (what does this mean?). If published, this will include your full peer review and any attached files.

Reviewer #1: No

Reviewer #2: No

Reviewer #3: No
---

## [Decision Letter · Decision Letter 1]

1 Jun 2021

Dear Dr. Cascante,

Thank you very much for submitting your manuscript "Integrating systemic and molecular levels to infer key drivers sustaining metabolic adaptations" for consideration at PLOS Computational Biology. As with all papers reviewed by the journal, your manuscript was reviewed by members of the editorial board and by several independent reviewers. The reviewers appreciated the attention to an important topic. Based on the reviews, we are likely to accept this manuscript for publication, providing that you modify the manuscript according to the review recommendations.

Sincerely,

Kiran Raosaheb Patil, Ph.D.

Deputy Editor

PLOS Computational Biology

Feilim Mac Gabhann

Editor-in-Chief

PLOS Computational Biology

[LINK]

Reviewer's Responses to Questions

**Comments to the Authors:**

Reviewer #1: the authors have completed a thorough and thoughtful revision. my previous comments have been addressed.

Reviewer #2: The authors have addressed all my comments, which were mainly concerning the structure of the manuscript. The revised version has substantially improved with regard to readability.

I have only a few additional minor suggestions:

- Figure 2 (network diagram) should come before Figure 1 (matrix coefficients of the network).

- The percentage gain metric should be explained somewhere, preferably before its first referral in figure 1.

- Panels B and C in figure 1 seem to be incorrectly labeled.

Reviewer #3: The authors have made a good effort to address all comments by the reviewers and have produced a much improved version of their paper, which can be accepted.

**Have the authors made all data and (if applicable) computational code underlying the findings in their manuscript fully available?**

Reviewer #1: Yes

Reviewer #2: None

Reviewer #3: Yes

PLOS authors have the option to publish the peer review history of their article (what does this mean?). If published, this will include your full peer review and any attached files.

Reviewer #1: No

Reviewer #2: No

Reviewer #3: No

Figure Files:

Data Requirements:

Reproducibility:

References:

---

## [Editor Report · Decision Letter 2]

1 Jul 2021

Dear Dr. Cascante,

We are pleased to inform you that your manuscript 'Integrating systemic and molecular levels to infer key drivers sustaining metabolic adaptations' has been provisionally accepted for publication in PLOS Computational Biology.

Best regards,

Kiran Raosaheb Patil, Ph.D.

Deputy Editor

PLOS Computational Biology

Feilim Mac Gabhann

Editor-in-Chief

PLOS Computational Biology

---

## [Editor Report · Acceptance letter]

19 Jul 2021

PCOMPBIOL-D-21-00015R2 

Integrating systemic and molecular levels to infer key drivers sustaining metabolic adaptations

Dear Dr Cascante,

I am pleased to inform you that your manuscript has been formally accepted for publication in PLOS Computational Biology. Your manuscript is now with our production department and you will be notified of the publication date in due course.

With kind regards,

Katalin Szabo
